# Adjuvant-dependent impact of inactivated SARS-CoV-2 vaccines during heterologous infection by a SARS-related coronavirus

Jacob A. Dillard [1,9], Sharon A. Taft-Benz[2,9], Audrey C. Knight[3], Elizabeth J. Anderson [4], Katia D. Pressey[4], Breantié Parotti[2], Sabian A. Martinez[2], Jennifer L. Diaz [1], Sanjay Sarkar[2], Emily A. Madden[1], Gabriela De la Cruz[5], Lily E. Adams[1], Kenneth H. Dinnon III[1], Sarah R. Leist [6], David R. Martinez[6], Alexandra Schäfer [6], John M. Powers [6], Boyd L. Yount Jr.[6], Izabella N. Castillo[1], Noah L. Morales[2], Jane Burdick[2], Mia Katrina D. Evangelista[5], Lauren M. Ralph[5], Nicholas C. Pankow[5], Colton L. Linnertz[2], Premkumar Lakshmanane[1], Stephanie A. Montgomery[3,7], Martin T. Ferris[2], Ralph S. Baric [1,6], Victoria K. Baxter [3,4,8] ✉ & Mark T. Heise [1,2] ✉

Whole virus-based inactivated SARS-CoV-2 vaccines adjuvanted with aluminum hydroxide have been critical to the COVID-19 pandemic response. Although these vaccines are protective against homologous coronavirus infection, the emergence of novel variants and the presence of large zoonotic reservoirs harboring novel heterologous coronaviruses provide significant opportunities for vaccine breakthrough, which raises the risk of adverse outcomes like vaccine-associated enhanced respiratory disease. Here, we use a female mouse model of coronavirus disease to evaluate inactivated vaccine performance against either homologous challenge with SARS-CoV-2 or heterologous challenge with a bat-derived coronavirus that represents a potential emerging disease threat. We show that inactivated SARS-CoV-2 vaccines adjuvanted with aluminum hydroxide can cause enhanced respiratory disease during heterologous infection, while use of an alternative adjuvant does not drive disease and promotes heterologous viral clearance. In this work, we highlight the impact of adjuvant selection on inactivated vaccine safety and efficacy against heterologous coronavirus infection.

Severe acute respiratory syndrome coronavirus 2 (SARS-CoV-2) has caused a global public health crisis resulting in nearly 7 million confirmed deaths and greater than 10 trillion dollars in economic losses[1,2]. In addition to the ongoing challenges caused by SARS-CoV-2, SARS-related coronaviruses (SARS-r-CoVs) continue to represent a major pandemic threat. Emergence events by zoonotic coronaviruses have occurred at least seven times throughout human history, with three highly pathogenic coronaviruses entering the human population from

[1]Department of Microbiology & Immunology, University of North Carolina at Chapel Hill, Chapel Hill, NC, USA. [2]Department of Genetics, University of North Carolina at Chapel Hill, Chapel Hill, NC, USA. [3]Department of Pathology & Laboratory Medicine, University of North Carolina at Chapel Hill, Chapel Hill, NC, USA. [4]Division of Comparative Medicine, University of North Carolina at Chapel Hill, Chapel Hill, NC, USA. [5]Pathology Services Core, University of North Carolina at Chapel Hill, Chapel Hill, NC, USA. [6]Department of Epidemiology, University of North Carolina at Chapel Hill, Chapel Hill, NC, USA. [7]Dallas Tissue Research, Farmers Branch, TX, USA. [8]Texas Biomedical Research Institute, San Antonio, TX, USA. [9]These authors contributed equally: Jacob A. Dillard, Sharon A. Taft-Benz. ✉e-mail: vkbaxter@txbiomed.org; mark_heisem@med.unc.edu

2003 to 2019 alone[3]. SARS-r-CoVs continue to circulate in zoonotic reservoirs and threaten to cause human infections[3–5]. Given the frequency of recent emergence events and the continuing threat posed by circulating SARS-r-CoVs, future zoonotic SARS-r-CoV epidemics are likely.

In the context of Coronavirus disease 19 (COVID-19), there has been an unprecedented effort devoted to the development, testing, and deployment of SARS-CoV-2 vaccines. To date, over 13.3 billion vaccine doses have been administered worldwide[6]. Among other important vaccine platforms, whole virus-based inactivated vaccines have had a major global impact on the COVID-19 pandemic. Inactivated vaccines are relatively simple to produce, lack special storage requirements, and are safe for immunocompromised individuals, making these vaccines attractive for widespread use[7]. Inactivated vaccines administered with the aluminum hydroxide adjuvant (Alum) accounted for approximately half of all COVID-19 vaccines (nearly 5 billion doses) administered by 2022[8]. Three inactivated COVID-19 vaccines (developed by Sinovac, Sinopharm, and Bharat Biotech) are approved for emergency use by the World Health Organization[9].

Inactivated vaccines provide moderate protection against symptomatic infection with significant and sustainable protection against severe disease and mortality[7,8,10–16]. However, neutralizing antibody titers induced by inactivated vaccines wane relatively quickly, and these vaccines are not highly effective against variants of concern (VOC) like B.1.617.2 (Delta) and Omicron subvariants[7,8,12,17]. This raises concerns about breakthrough infections in vaccinated individuals, particularly in individuals who do not mount strong immune responses to vaccines, including those 65 years of age and older.

Such breakthrough infections due to vaccine failure are sometimes associated with vaccine-associated enhanced respiratory disease (VAERD), an outcome that has been observed historically with the formalin-inactivated respiratory syncytial virus (RSV) and formalin-inactivated measles virus vaccines[18–27]. Vaccine-associated pathology was reported in several preclinical studies of SARS-CoV and MERS-CoV vaccines, including inactivated vaccines, replicon-vectored vaccines, and recombinant subunit spike protein vaccines[4,5,28–41]. Vaccine-induced pathology following homologous or heterologous viral infection was characterized by type 2 immunopathology, including pulmonary eosinophil infiltration and upregulation of type 2 cytokines. The majority of these studies were performed in BALB/c mice; however, vaccine-enhanced immunopathology was also reported in C57BL/6 mice, ferrets, and non-human primates. Additionally, three recent studies have reported VAERD in rodent models involving SARS-CoV-2 vaccines[42–44]. DiPiazza and Leist et al. found that inactivated SARS-CoV-2 vaccines induce suboptimal immune responses, including weak neutralizing antibodies, and can cause type 2-associated immunopathology in BALB/cJ mice following SARS-CoV-2 infection[42]. Ebenig et al. reported type 2 immunopathology in SARS-CoV-2-infected hamsters that were previously vaccinated with a suboptimal spike protein vaccine[43]. Iwata-Yoshikawa and Shiwa et al. observed adjuvant-dependent outcomes using a recombinant spike protein vaccine[44].

Given the rising threat of vaccine breakthrough by current VOC and the potential for future SARS-r-CoV epidemics, combined with the risk of VAERD in the context of vaccine failure, we evaluated the impact of adjuvants on the safety and efficacy of an inactivated SARS-CoV-2 vaccine (iCoV2), using established mouse models of homologous and heterologous SARS-r-CoV respiratory infection.

Here, we show that the safety and efficacy of iCoV2 during heterologous infection by a SARS-r-CoV is adjuvant-dependent. iCoV2 formulated with Alum (iCoV2 + Alum) protects against infection by a homologous virus and a VOC (B.1.351), but results in type 2 VAERD, including enhanced clinical disease, in mice during infection by a heterologous virus, Rs-SHC014-CoV (SHC014). iCoV2 + Alum vaccination also fails to control SHC014 replication, causes delayed SHC014 clearance, and impairs respiratory function in a subset of vaccinated animals. To evaluate the importance of adjuvant choice in determining vaccine safety and efficacy during heterologous infection, we administered iCoV2 with RIBI, a research-grade adjuvant also referred to as Sigma Adjuvant System. RIBI is an oil-in-water emulsion (2% squalene and Tween 80) containing monophosphoryl lipid A, a non-toxic analog of lipopolysaccharide that stimulates toll-like receptor 4, and synthetic trehalose dicorynomycolate, a low-toxicity derivative of mycobacterial cord factor trehalose-6,6-dimycolate that is thought to stimulate a C-type lectin expressed by macrophages known as the Mincle receptor[45–48]. In strong contrast to iCoV2 + Alum, when administered with RIBI, iCoV2 cross-protects against SHC014 without causing VAERD. We also observed that secondary boost vaccination with a heterologous vaccine lacking Alum partially reduces VAERD. Lastly, we find that CD4+ T helper (TH) cells play a major role in driving VAERD. In summary, our findings (i) indicate that the safety and efficacy of an inactivated SARS-CoV-2 vaccine during heterologous SARS-r-CoV infection are adjuvant-dependent, (ii) demonstrate the utility of evaluating coronavirus vaccines against heterologous viruses to identify potential safety issues during preclinical development, (iii) underscore the need to preempt vaccine breakthrough and VAERD with universal coronavirus vaccines, and (iv) suggest the need for increased surveillance to identify VAERD in humans receiving Alum-based inactivated coronavirus vaccines.

## Results

### Inactivated SARS-CoV-2 vaccine protects mice against SARS-CoV-2

To evaluate the impact of adjuvant formulation on the safety and efficacy of iCoV2 during homologous or heterologous viral challenge, we used established mouse models of vaccination and SARS-CoV-2 and SARS-r-CoV challenge[4,49,50]. BALB/c mice were vaccinated with iCoV2, which was derived from an infectious clone of an early pandemic isolate of SARS-CoV-2, the D614 strain based on the WA1 sequence[51]. iCoV2 was administered with either the Alum adjuvant (iCoV2 + Alum) or RIBI (Sigma Adjuvant System, iCoV2 + RIBI), a research-grade adjuvant reported to induce type 1-biased immune responses[45,46]. Mock-vaccinated mice received an irrelevant viral antigen plus Alum (inactivated influenza virus, iFLU + Alum). Both inactivated SARS-CoV-2 vaccine formulations were highly immunogenic, as demonstrated by the induction of robust neutralizing antibodies against the homologous SARS-CoV-2 strain, although we did detect approximately threefold higher neutralization titers induced by iCoV2 + RIBI compared to iCoV2 + Alum ($\log_{10}[2.838]$ vs. $\log_{10}[2.363]$, respectively; $p = 0.0054$) (Fig. 1a). Next, upon challenge with pathogenic mouse-adapted SARS-CoV-2 (MA10)[49,50], mice vaccinated with either iCoV2 formulation were protected from the clinical disease compared to mock-vaccinated controls (Fig. 1b, c) and exhibited undetectable pulmonary viral titers at 5 DPI (Fig. 1d). Both vaccines also significantly reduced pulmonary pathology, as measured by acute lung injury (ALI) and diffuse alveolar damage (DAD) (Fig. 1e,f)[49,50,52–55]. Notably, iCoV2 + RIBI provided more complete protection than the Alum vaccine from ALI and DAD, although these differences were not statistically significant. Lastly, consistent with Dipiazza et al.[42], we observed adjuvant-dependent pulmonary eosinophil infiltration and type 2 cytokine upregulation in iCoV2 + Alum-vaccinated mice (Fig. 1g, h and Supplementary Fig. 1a). Thus, iCoV2 is protective against pathogenic homologous virus challenge, consistent with results from inactivated COVID-19 vaccines in humans[8], but RIBI promotes improved protection from pathology, including avoidance of type 2 inflammation.

Because inactivated COVID-19 vaccines exhibit reduced efficacy against VOC in humans[8,12,17], we further evaluated the immunogenicity of iCoV2 against a panel of VOC using an established surrogate neutralization assay measuring inhibition of binding between human angiotensin-converting enzyme 2 (ACE2) and SARS-CoV-2 spike proteins. Consistent with the real-world performance of inactivated SARS-

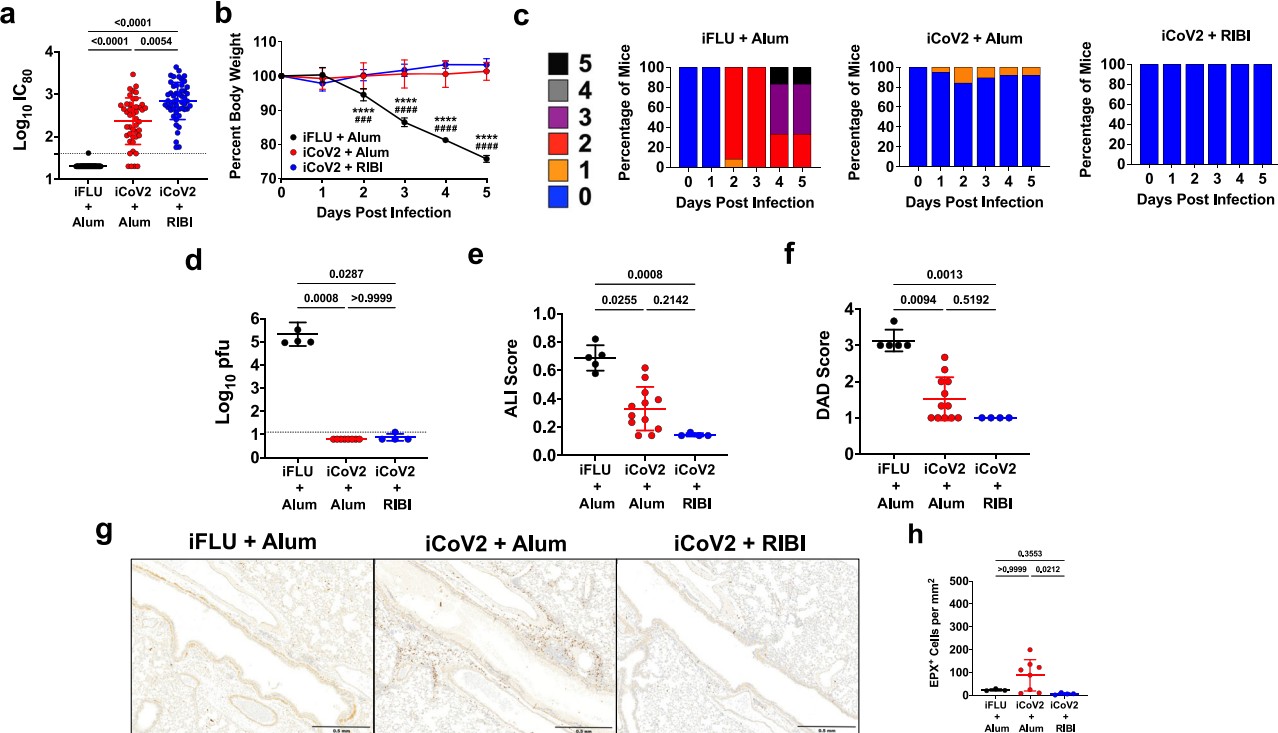

**Fig. 1 | Inactivated vaccine protects mice against SARS-CoV-2 but causes adjuvant-dependent type 2 inflammation. a** Post-boost serum-neutralizing antibody titers against SARS-CoV-2 (D614G) in mice vaccinated with inactivated influenza virus adjuvanted with aluminum hydroxide (iFLU + Alum, $n = 49$), inactivated SARS-CoV-2 adjuvanted with aluminum hydroxide (iCoV2 + Alum, $n = 45$), or inactivated SARS-CoV-2 adjuvanted with Sigma Adjuvant System adjuvant (iCoV2 + RIBI, $n = 50$); $IC_{80}$ = 80% inhibitory concentration. **b, c** Body weight change (**b**) and clinical scores (**c**) in iFLU + Alum ($n = 12$), iCoV2 + Alum ($n = 19$), or iCoV2 + RIBI ($n = 4$) vaccinated mice following challenge with SARS-CoV-2-MA10. **d** Pulmonary viral titers at 5 DPI following SARS-CoV-2-MA10 infection in iFLU + Alum ($n = 5$), iCoV2 + Alum ($n = 8$), or iCoV2 + RIBI ($n = 4$) vaccinated mice; pfu plaque-forming units. **e, f** Acute lung injury (ALI) (**e**) and diffuse alveolar damage (DAD) (**f**) scores in hematoxylin and eosin (H&E)-stained lungs at 5 DPI following SARS-CoV-2-MA10 infection in iFLU + Alum ($n = 5$), iCoV2 + Alum ($n = 12$), or iCoV2 + RIBI ($n = 4$) vaccinated mice. **g, h** Representative photomicrographs (**g**) (scale bar = 0.5 mm) and quantification (**h**) of pulmonary eosinophils immunohistochemically labeled for

eosinophil peroxidase (EPX, brown cells) at 5 DPI following SARS-CoV-2-MA10 infection in iFLU + Alum ($n = 3$), iCoV2 + Alum ($n = 8$), or iCoV2 + RIBI ($n = 4$) vaccinated mice. **a, d–f, h** Individual data points represent independent biological replicates; solid horizontal lines and error bars represent group means ± standard deviation (SD); data analyzed by Kruskal–Wallis test with Dunn's multiple comparisons; solid horizontal lines above data represent pairwise comparisons with $p$ values; **a, d** Dotted line represents assay limit of detection. **b** Results reported as mean ± SD; ****$p < 0.0001$, iFLU + Alum versus iCoV2 + Alum, ###$p < 0.001$, ####$p < 0.0001$, iFLU + Alum versus iCoV2 + RIBI, two-way ANOVA with Tukey's multiple comparisons. **c** Clinical scoring system: 0 = normal (blue), 1 = piloerection (orange), 2 = piloerection + kyphosis (red), 3 = piloerection, kyphosis, and reduced movement (purple), 4 = markedly reduced movement and/or labored breathing (gray), and 5 = moribund, dead or euthanized (black). Data presented as combined results from one (**b, c**), two (**e, f, h**), three (**d**), or four (**a**) independent animal experiments.

CoV-2 vaccines, both iCoV2 formulations elicited robust neutralizing activity against early pandemic VOC, including B.1.351 (Beta) and B.1.617.2 (Delta) (Supplementary Fig. 2a–e), but little neutralizing activity against Omicron subvariants (Supplementary Fig. 2f–j).

We also evaluated vaccine efficacy against challenge with B.1.351 using a mouse-adapted SARS-CoV-2 expressing the B.1.351 spike protein (MA10-B.1.351)[56]. Similar to homologous challenge, iCoV2 induced neutralizing antibodies, promoted control of viral replication, and prevented severe clinical disease and pathology in mice challenged with MA10-B.1.351 (Supplementary Fig. 3a–f). Notably, we observed mild transient weight loss in vaccinated mice (Supplementary Fig. 3b) and mild clinical signs specifically in iCoV2 + Alum-vaccinated mice (Supplementary Fig. 3c). Furthermore, while iCoV2 + RIBI provided protection against respiratory pathology, iCoV2 + Alum conferred incomplete protection that was not significantly different from controls (Supplementary Fig. 3e, f). Lastly, like the homologous challenge results, we again observed type 2 inflammation specifically associated with iCoV2 + Alum, although the magnitude of type 2 cytokine upregulation was lower than that observed during the homologous challenge (Supplementary Fig. 1b), which may reflect lower viral loads in

this challenge model compared to the homologous challenge model (Fig. 1d and Supplementary Fig. 3d; iFLU + Alum group).

## Inactivated SARS-CoV-2 vaccine causes adjuvant-dependent enhanced disease during infection by a SARS-related coronavirus

Preexisting vaccine-induced SARS-CoV-2 immune memory will likely impact potential future SARS-r-CoV epidemics. Based on the performance of iCoV2 against VOC, we expected that iCoV2 would fail to elicit protective immunity against a heterologous pre-emergent SARS-r-CoV. To this end, we measured serum neutralization against SHC014, a clade 3 sarbecovirus that can bind to human ACE2 with high affinity and replicate in human airway epithelial cells, making it a potential emerging disease threat[4]. Using mRNA vaccine sera against SARS-CoV-2 D614G, previous studies show an approximately 100-fold reduction in neutralizing titers to SHC014[56]. Under conditions in which iCoV2 + Alum did not induce detectable neutralizing antibodies above background, iCoV2 + RIBI elicited a threefold increase in cross-neutralizing antibody titers against SHC014 from baseline. These responses were significantly different from both the control and iCoV2 + Alum groups

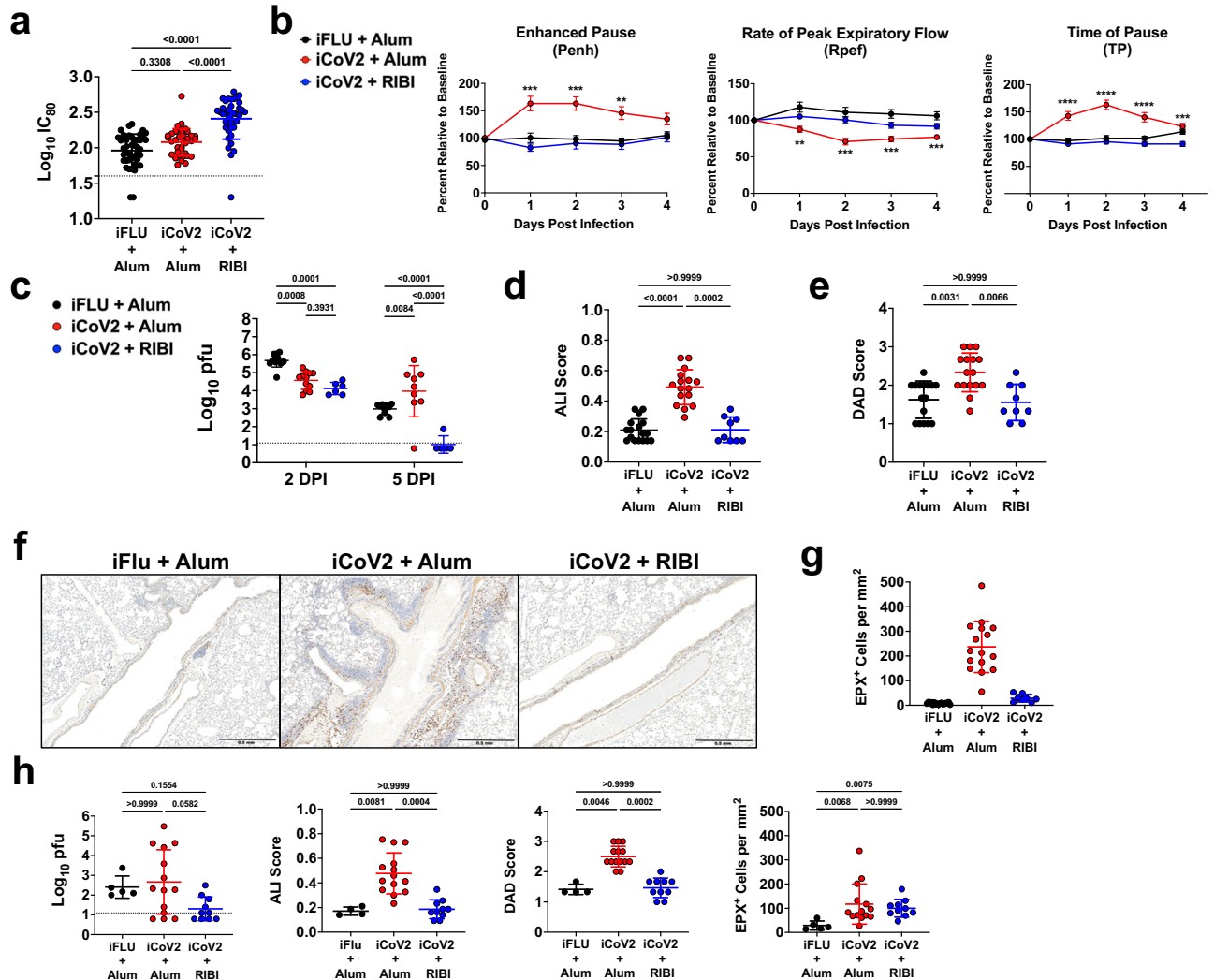

**Fig. 2 | Inactivated SARS-CoV-2 vaccine causes enhanced disease during infection by a SARS-related coronavirus. a** Post-boost serum-neutralizing antibody titers against Rs-SHC014-CoV in mice vaccinated with inactivated influenza virus adjuvanted with aluminum hydroxide (iFLU + Alum, $n = 43$), inactivated SARS-CoV-2 adjuvanted with aluminum hydroxide (iCoV2 + Alum, $n = 36$), or inactivated SARS-CoV-2 adjuvanted with Sigma Adjuvant System adjuvant (iCoV2 + RIBI, $n = 39$); $IC_{80}$ = 80% inhibitory concentration. **b** Pulmonary function measured by whole-body plethysmography in iFLU + Alum ($n = 12$), iCoV2 + Alum ($n = 24$), or iCoV2 + RIBI ($n = 12$) vaccinated mice following challenge with Rs-SHC014-CoV 4 weeks post-boost; results reported as mean ± standard error of the mean; **$p < 0.01$, **$p < 0.001$, ****$p < 0.0001$, iFLU + Alum versus iCoV2 + Alum, two-way ANOVA with Geisser-Greenhouse correction and Tukey's multiple comparisons. **c** Pulmonary viral titers following Rs-SHC014-CoV infection in iFLU + Alum (2 DPI $n = 12$, 5 DPI $n = 10$), iCoV2 + Alum (2 DPI $n = 12$, 5 DPI $n = 9$), or iCoV2 + RIBI (2 DPI $n = 6$, 5 DPI $n = 5$) -vaccinated mice; pfu plaque-forming units. **d**, **e** Acute lung injury (ALI) (**d**) and diffuse alveolar damage (DAD) (**e**) scores in hematoxylin and eosin

(H&E)-stained lungs at 5 DPI following Rs-SHC014-CoV infection in iFLU + Alum (ALI $n = 17$, DAD $n = 16$), iCoV2 + Alum ($n = 16$), or iCoV2 + RIBI ($n = 9$) vaccinated mice. **f**, **g** Representative photomicrographs (**f**) (scale bar = 0.5 mm) and quantification (**g**) of pulmonary eosinophils immunohistochemically labeled for eosinophil peroxidase (EPX, brown cells) at 5 DPI following Rs-SHC014-CoV infection in iFLU + Alum ($n = 17$), iCoV2 + Alum ($n = 16$), or iCoV2 + RIBI ($n = 8$) vaccinated mice. **h** Viral lung titers (left panel), ALI scores (second from left panel), DAD scores (second from right panel), and EPX$^+$ cells (right panel) at 5 DPI in mice vaccinated with iFLU + Alum (titer and EPX $n = 5$, ALI and DAD $n = 4$), iCoV2 + Alum ($n = 14$), or iCoV2 + RIBI ($n = 10$) and challenged with Rs-SHC014-CoV 10.5 months post-boost. **a**, **c**–**e**, **g**, **h** Individual data points represent independent biological replicates taken from discrete samples; horizontal lines and error bars represent group means ± standard deviation; data analyzed by Kruskal–Wallis test with Dunn's multiple comparisons. **a**, **c**, **h** Dotted line represents the assay limit of detection. Data presented as combined results from one (**c**, **h**), two (**b**), three (**d**, **e**, **g**), or four (**a**) independent animal experiments.

(Fig. 2a), suggesting adjuvant-dependent effects on heterologous neutralizing antibody responses.

Due to the poor cross-neutralization elicited by iCoV2 + Alum against SHC014, which increases the risk of vaccine breakthrough, we next assessed vaccine-mediated protection against the SHC014 challenge. SHC014 can replicate to high levels in murine respiratory tissue but causes minimal pathology and no overt clinical disease in naïve mice[4], thus providing an optimal model for detecting VAERD. Consistent with this prior work, SHC014 did not cause overt clinical disease (Supplementary Fig. 4), alter respiratory function, or induce pathology in control mice (Fig. 2). However, iCoV2 + Alum-vaccinated mice

exhibited impaired respiratory function as measured by whole-body plethysmography (WBP), including altered Enhanced Pause (Penh), Rate of Peak Expiratory Flow (Rpef), and Time of Pause (TP), compared to controls (Fig. 2b). Importantly, iCoV2 + RIBI vaccination had no adverse effect on any respiratory function measure. Both vaccine formulations caused modest reductions in viral load at 2 DPI, demonstrating a degree of cross-protection (Fig. 2c). However, compared to controls, iCoV2 + Alum-vaccinated mice exhibited significantly higher viral loads at 5 DPI, with many mice exhibiting titers equivalent to those seen at 2 DPI. In contrast, iCoV2 + RIBI promoted robust clearance by 5 DPI. These results were corroborated by

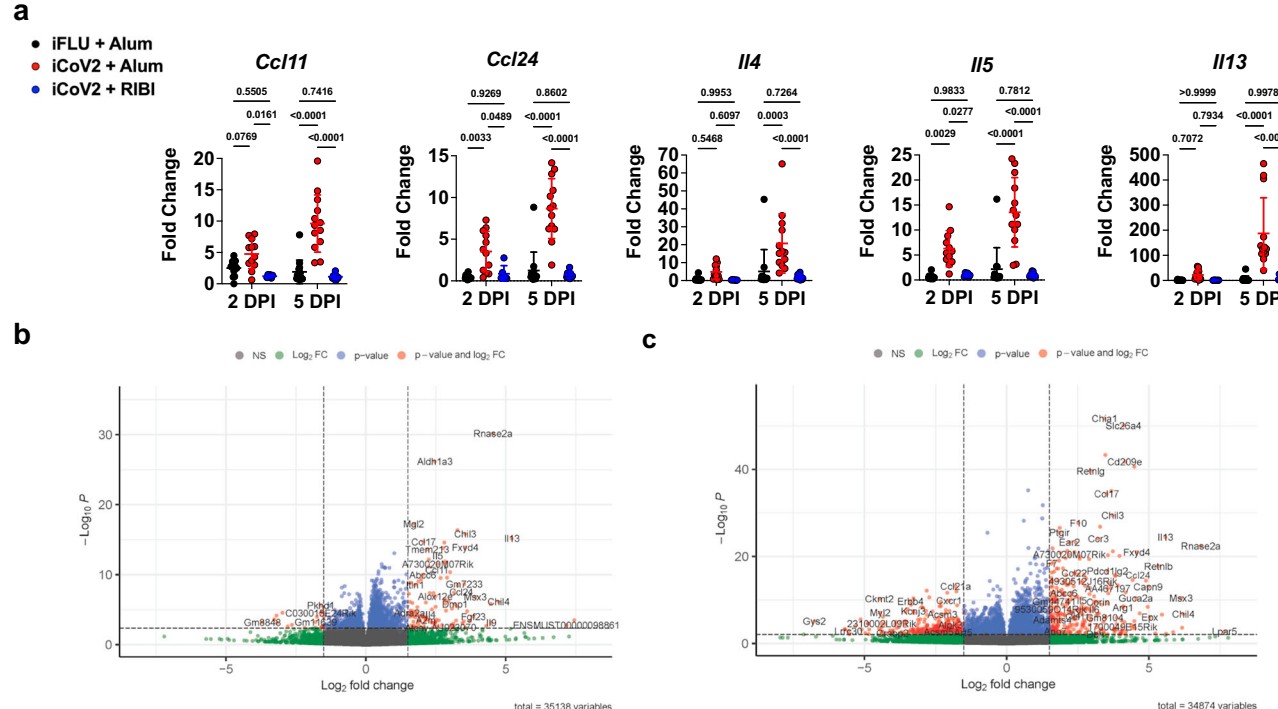

**Fig. 3 | Vaccine adjuvants promote divergent immune gene expression patterns during heterologous infection. a** Type 2 cytokine gene expression (*Ccl11* C-C motif chemokine 11, *Ccl24* C-C motif chemokine 24, *Il4* interleukin 4, *Il5* interleukin 5, *Il13* interleukin 13) normalized to *Gapdh* expression at 2- and 5 days post-infection (DPI) with Rs-SHC014-CoV in mice vaccinated with inactivated influenza virus adjuvanted with aluminum hydroxide (iFLU + Alum), inactivated SARS-CoV-2 adjuvanted with aluminum hydroxide (iCoV2 + Alum), or inactivated SARS-CoV-2 adjuvanted with Sigma Adjuvant System adjuvant (iCoV2 + RIBI); *Ccl11, Ccl24* 2 DPI – iFLU + Alum (*n* = 12), iCoV2 + Alum (*n* = 12), iCoV2 + RIBI (*n* = 6); *Ccl11, Ccl24* 5 DPI – iFLU + Alum (*n* = 14), iCoV2 + Alum (*n* = 13), iCoV2 + RIBI (*n* = 9); *Il4, Il5, Il13* 2 DPI – iFLU + Alum (*n* = 12), iCoV2 + Alum (*n* = 12), iCoV2 + RIBI (*n* = 6); *Il4, Il5, Il13* 5 DPI – iFLU + Alum (*n* = 13), iCoV2 + Alum (*n* = 13), iCoV2 + RIBI (*n* = 9); individual data points represent independent biological replicates; results presented from one animal experiment and analyzed by Kruskal–Wallis test with Dunn's multiple comparisons correction; solid horizontal lines and error bars overlaying data

represent group means ± standard deviation; solid horizontal lines above data represent pairwise comparisons with *p* values. **b, c** Volcano plots showing differential expression by RNA-Seq between iCoV2 + Alum relative to iCoV2 + RIBI at 2 DPI (**b**) and 5 DPI (**c**) following infection with Rs-SHC014-CoV. Fold change (FC) is shown along the X-axis (with 1.5 log$_2$ FC thresholds represented by vertical dashed lines) and significance along the Y-axis (with false discovery rate-adjusted *q* < 0.05 thresholds represented by horizontal dashed lines). Key genes are highlighted in the upper right (iCoV2 + Alum expression > iCoV2 + RIBI) and upper left (iCoV2 + Alum expression < iCoV2 + RIBI) quadrants. 2 DPI (*n* = 12) and 5 DPI (*n* = 9) for iCoV2 + Alum, and 2 DPI (*n* = 6) and 5 DPI (*n* = 5) for iCoV2 + RIBI; results presented from one animal experiment and analysis described in detail in RNA Sequencing (RNA-Seq) in Methods. Source data are provided as a Source Data file. RNA Sequencing raw .fastq data files are submitted to the Sequence Read Archive (SRA) database under BioProject ID PRJNA1022427.

immunohistochemical staining of viral nucleocapsid antigen (Supplementary Fig. 5a). In addition to causing impaired respiratory function and delayed viral clearance, iCoV2 + Alum vaccination caused increased pathology at 5 DPI, while iCoV2 + RIBI-vaccinated mice showed no signs of respiratory pathology (Fig. 2d, e). Furthermore, iCoV2 + Alum-vaccinated mice specifically exhibited type 2 inflammation during SHC014 infection, including increased pulmonary eosinophil infiltration by 5 DPI and type 2 cytokine expression at both 2 and 5 DPI (Figs. 2f, g, 3a). Pathological analysis also revealed additional inflammatory signs unique to iCoV2 + Alum-vaccinated mice: enhanced infiltration of CD4$^+$ cells (likely predominantly T$_H$ cells) at 5 DPI, C3 complement protein deposition at 2 and 5 DPI, Arginase$^+$ cell infiltration at 2 and 5 DPI, and mucus production at 5 DPI (Supplementary Fig. 5b–f). These results clearly demonstrate that in the context of heterologous infection, vaccination with iCoV2 + Alum not only fails to protect against SHC014 replication but actually predisposes animals to enhanced virus-induced disease and delayed viral clearance. In contrast, the use of the RIBI adjuvant promotes viral clearance and is not associated with exacerbated disease.

We next assessed the durability of either the adverse effects of iCoV2 + Alum or the protective effects of iCoV2 + RIBI by challenging mice up to 10.5 months post-boost. We observed that iCoV2 + Alum caused delayed viral clearance by 5 DPI in a subset of mice challenged

at all time points (Fig. 2h and Supplementary Fig. 6a), while iCoV2 + RIBI appeared to promote clearance as late as 10.5 months post-boost. WBP revealed signs of impaired respiratory function in iCoV2 + Alum-vaccinated mice challenged up to 9 months post-boost (Supplementary Fig. 6b). We also observed adjuvant-dependent exacerbated pathology and type 2 inflammation in mice challenged through the duration of the study (Fig. 2h and Supplementary Fig. 6c–e). Therefore, susceptibility to VAERD seen in iCoV2 + Alum-vaccinated animals is highly durable. Of note, we did observe mild pulmonary eosinophilic infiltration in the iCoV2 + RIBI group challenged 10.5 months post-boost vaccination (Fig. 2h), possibly due to waning vaccine protection and/or age-associated changes in the immune response.

Because we observed high variability in viral loads in the iCoV2 + Alum group infected at 10.5 months post-boost (Fig. 2h), we further analyzed this group to assess for correlation between lung titers and other disease parameters, including eosinophil infiltration, CD4$^+$ cell infiltration, ALI, and DAD. We found a significant correlation between lung titers and ALI ($r$ = 0.67, $p$ = 0.011) (Supplementary Fig. 7). Although not statistically significant, we observed that lung titers may have correlated moderately with CD4$^+$ cell infiltration ($r$ = 0.50, $p$ = 0.071) and DAD ($r$ = 0.48, $p$ = 0.085). We found no evidence of a significant correlation between lung titers and eosinophil infiltration ($r$ = 0.27, $p$ = 0.355). In fact, eosinophil infiltration did not correlate

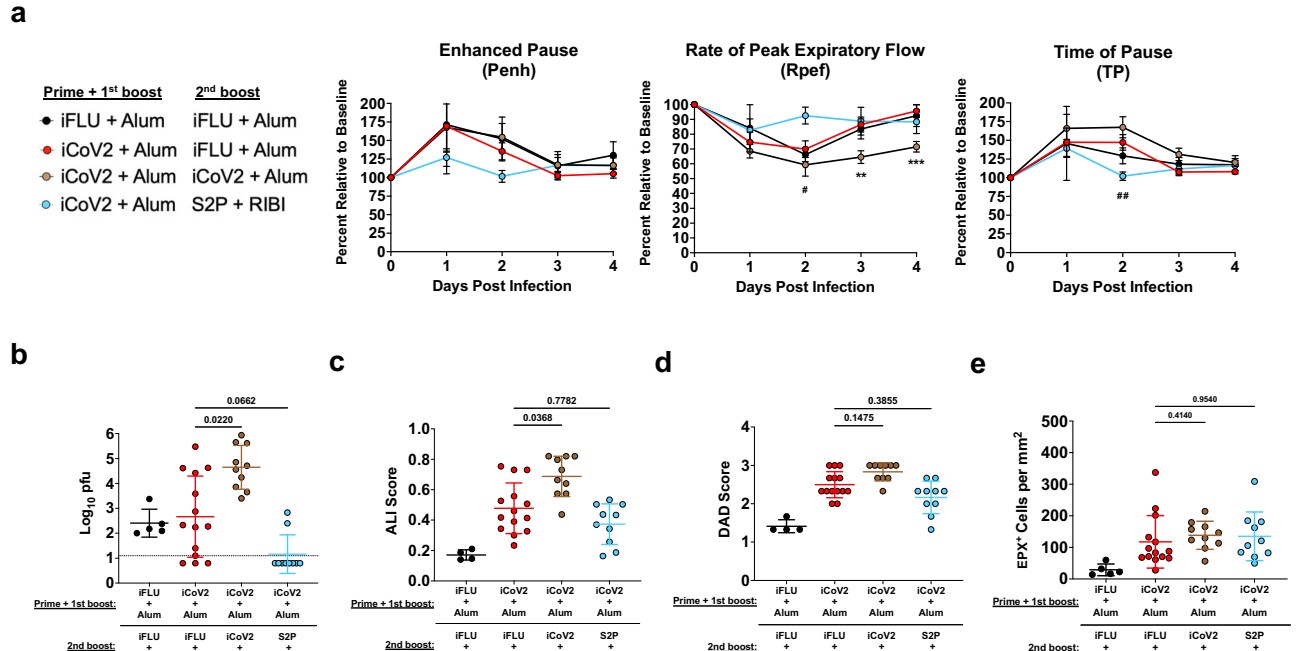

**Fig. 4 | Alternative boost vaccination partially reduces vaccine-enhanced disease. a** Pulmonary function measured by whole-body plethysmography following Rs-SHC014-CoV challenge in mice vaccinated with inactivated influenza virus adjuvanted with aluminum hydroxide (iFLU + Alum) initially followed by a second boost vaccination with iFLU + Alum 9.5 months post-first boost (*n* = 5) or vaccinated with inactivated SARS-CoV-2 adjuvanted with aluminum hydroxide (iCoV2 + Alum) initially followed by a second boost vaccination with iFLU + Alum (*n* = 14), iCoV2 + Alum (*n* = 10), or stabilized spike protein adjuvanted with Sigma Adjuvant System adjuvant (S2P + RIBI, *n* = 9) 9.5 months post-first boost; results represented by one experiment and reported as group mean ± standard error of the mean; **$p < 0.01$, ***$p < 0.001$, iCoV2 + Alum – iFLU + Alum versus iCoV2 + Alum – iCoV2 + Alum, #$p < 0.05$, ##$p < 0.01$, iCoV2 + Alum – iFLU + Alum versus iCoV2 + Alum – S2P + RIBI, two-way ANOVA with Geisser-Greenhouse correction and Dunnett's multiple comparisons correction. **b** Pulmonary viral titers at 5 days post-infection (DPI) following Rs-SHC014-CoV infection; dotted line represents

assay limit of detection; pfu plaque forming units. **c, d** Acute lung injury (ALI) (**c**) and diffuse alveolar damage (DAD) (**d**) scores in hematoxylin and eosin (H&E)-stained lungs at 5 DPI following Rs-SHC014-CoV infection. **e** Quantification of pulmonary eosinophils immunohistochemically labeled for eosinophil peroxidase (EPX, brown cells) at 5 DPI following Rs-SHC014-CoV infection. **b–e** Individual data points represent independent biological replicates; solid horizontal lines and error bars overlaying data represent group mean ± standard deviation; data analyzed by Kruskal–Wallis test with Dunn's multiple comparisons correction; solid horizontal lines above data represent pairwise comparisons with *p* values; iFLU + Alum – iFLU + Alum (*n* = 5), iCoV2 + Alum – iFLU + Alum (*n* = 14), iCoV2 + Alum – iCoV2 + Alum (*n* = 10), iCoV2 + Alum – S2P + RIBI (*n* = 10). iCoV2 + Alum – iFLU + Alum control group data are repeated from Fig. 2 and served as the iCoV2 + Alum group for the final 10.5-month post-boost vaccination (Fig. 2h) and the control group for comparison to later secondary boost vaccination (Fig. 4).

significantly with any other parameters, which may be due in part to the fact that we observed a relatively low magnitude of eosinophil infiltration at this 10.5-month time point. We also found that CD4[+] cell infiltration correlated significantly with ALI ($r = 0.72$, $p = 0.005$) and DAD ($r = 0.73$, $p = 0.004$). As expected, we also saw a strong correlation between ALI and DAD ($r = 0.83$, p = 3.68E-4).

## Adjuvants promote divergent gene expression patterns during heterologous infection

To systematically evaluate how iCoV2 + Alum and iCoV2 + RIBI alter the pulmonary immune environment during SHC014 infection, we conducted RNA Sequencing (RNA-Seq) and analysis of differentially expressed genes (DEG). We found 2042 DEG at 2 DPI, 4349 DEG at 5 DPI, and 738 of these DEG at both time points (at a Bonferroni-corrected $p < 0.05$). At 2 DPI, 1529 genes were significantly upregulated, and 513 were significantly downregulated, in iCoV2 + Alum mice relative to iCoV2 + RIBI counterparts (Fig. 3b). At 5 DPI, 2158 genes were significantly upregulated, and 2191 were significantly downregulated, in iCoV2 + Alum relative to iCoV2 + RIBI (Fig. 3c).

Given several of the genes that were significantly upregulated in iCoV2 + Alum-vaccinated mice were associated with type 2 inflammation and eosinophil recruitment (e.g., *Il4*, *Il5*, *Il13*, *Ccl11*, and *Ccl24*), we sought to identify biological processes that were differentially functioning between the two vaccines. Using Gene Ontology (GO) enrichment analysis of genes that were (i) significantly differentially

expressed and (ii) had a $\log_2$ fold change ≥ 1.5, we found that chemokine and cytokine activity, and specifically CCR chemokine receptor binding functions, were significantly enriched in the iCoV2 + Alum group at 2 DPI (Supplementary Table 1). Similarly, the predominant pathways upregulated in the iCoV2 + Alum group at 5 DPI continued to include chemokine and cytokine signaling. In contrast, the upregulated pathways in iCoV2 + RIBI at 5 DPI showed a variety of normal processes (e.g., ion transport activities and cytoskeletal binding), consistent with a resolution of infection in these animals (Supplementary Table 1).

## Alternative boost vaccination partially reduces vaccine-enhanced disease

Given the large number of people vaccinated with Alum-adjuvanted inactivated COVID-19 vaccines, the risk of individuals developing VAERD upon exposure to a newly emerging heterologous coronavirus represents a potential public health risk. Therefore, we tested whether this risk could be ameliorated by rebooting iCoV2 + Alum-vaccinated mice with a RIBI-adjuvanted pre-fusion stabilized recombinant spike protein vaccine (S2P + RIBI). For comparison, we also included a group that received a third dose of iCoV2 + Alum. Compared to iCoV2 + Alum controls (iFLU + Alum secondary boost), a third dose of iCoV2 + Alum promoted impaired respiratory function, indicated by decreased Rpef at 3 and 4 DPI (Fig. 4a), more severe delayed viral clearance (Fig. 4b), and exacerbated pathology (Fig. 4c–e and Supplementary Fig. 8)

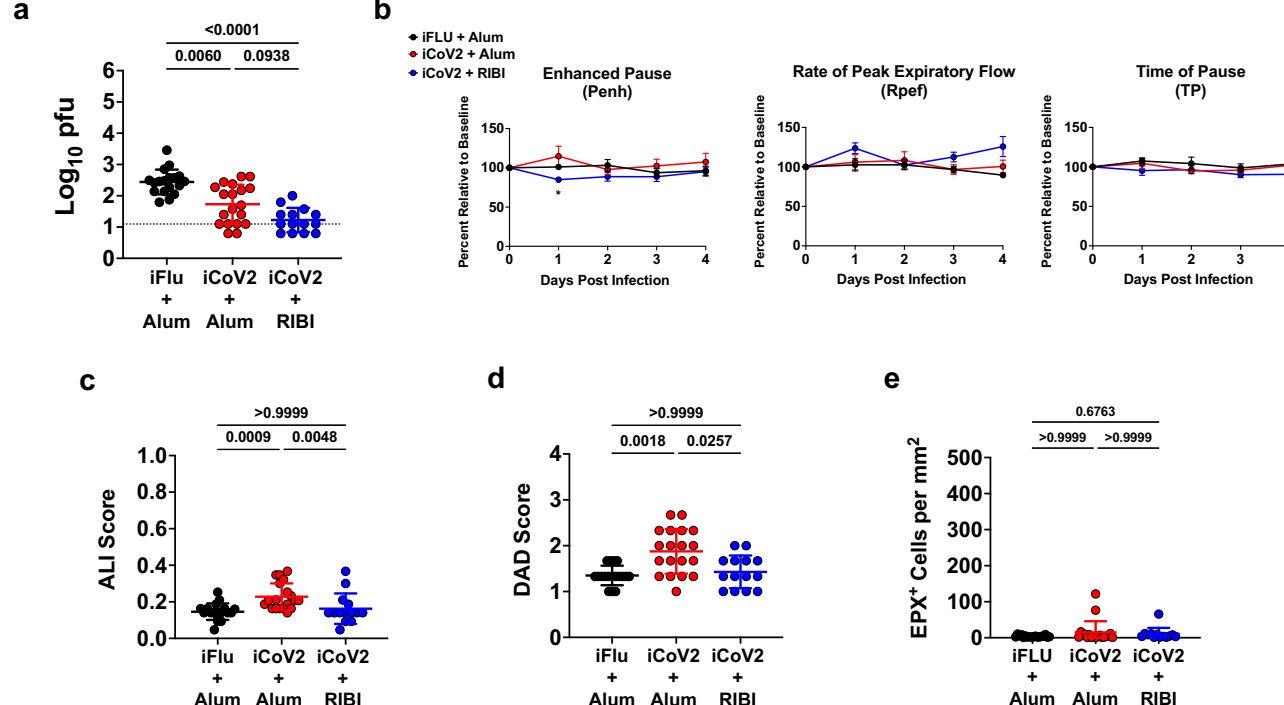

**Fig. 5 | Vaccine immune serum promotes cross-protection with modest pathology during heterologous infection. a** Pulmonary viral titers at 5 days post-infection (DPI) following Rs-SHC014-CoV infection in naïve mice that received a passive serum transfer from mice vaccinated with inactivated influenza virus adjuvanted with aluminum hydroxide (iFLU + Alum, $n = 18$), inactivated SARS-CoV-2 adjuvanted with aluminum hydroxide (iCoV2 + Alum, $n = 19$), or inactivated SARS-CoV-2 adjuvanted with Sigma Adjuvant System adjuvant (iCoV2 + RIBI, $n = 14$); dotted line represents assay limit of detection; pfu plaque forming units. **b** Pulmonary function measured by whole-body plethysmography following challenge with Rs-SHC014-CoV in naïve mice that received a passive serum transfer from mice vaccinated with iFLU + Alum ($n = 6$), iCoV2 + Alum ($n = 6$), or iCoV2 + RIBI ($n = 6$); results reported as group mean ± standard error of the mean; *$p < 0.05$, iFLU + Alum versus iCoV2 + Alum, two-way ANOVA with Geisser-Greenhouse correction and Tukey's multiple comparisons correction. **c, d** Acute lung injury (ALI)

(c) and diffuse alveolar damage (DAD) (**d**) scores in hematoxylin and eosin (H&E)-stained lungs at 5 DPI following Rs-SHC014-CoV infection in naïve mice that received a passive serum transfer from mice vaccinated with iFLU + Alum ($n = 18$), iCoV2 + Alum ($n = 19$), or iCoV2 + RIBI ($n = 14$). **e** Quantification of pulmonary eosinophils immunohistochemically labeled for eosinophil peroxidase (EPX, brown cells) at 5 DPI following Rs-SHC014-CoV infection in naïve mice that received a passive serum transfer from mice vaccinated with iFLU + Alum ($n = 18$), iCoV2 + Alum ($n = 19$), or iCoV2 + RIBI ($n = 13$). Individual data points represent independent biological replicates; solid lines and error bars represent group mean ± standard deviation; data analyzed by Kruskal–Wallis test with Dunn's multiple comparisons correction; solid horizontal lines above data represent pairwise comparisons with $p$ values; data presented as combined results from one (**b**) or two (**a, c, d, e**) independent animal experiments.

compared to iCoV2 + Alum controls. In contrast, secondary boost vaccination with S2P + RIBI modestly protected against impaired respiratory function, indicated by the maintenance of Rpef and TP at 2 DPI, and promoted improved viral clearance, with 8 of 10 mice exhibiting no detectable virus at 5 DPI (Fig. 4a, b). However, reboost with S2P + RIBI did not provide significant protection against increased pathology, eosinophil infiltration, or mucus production compared to iCoV2 + Alum controls (Fig. 4c–e and Supplementary Fig. 8). Therefore, alternative boost vaccination with S2P + RIBI partially protects from impaired respiratory function and delayed viral clearance, while not resolving other pathologic pulmonary responses associated with iCoV2 + Alum. To elucidate the protective mechanism(s) of S2P + RIBI, we analyzed serum-neutralizing antibody titers against SARS-CoV-2 and SHC014. Alternative boost with S2P + RIBI induced high homologous neutralizing titers. However, due to high levels of background neutralization activity against SHC014 in the older mice, we were unable to detect cross-neutralizing titers against SHC014 above background levels (Supplementary Fig. 9).

## CD4[+] T helper cells drive vaccine-enhanced disease

The strong type 2 inflammatory profile observed with iCoV2 + Alum-induced VAERD (Figs. 2f, g, 3) is similar to the immunopathology caused by formalin-inactivated RSV vaccination, which is reported to be mediated by $T_H$ type 2 ($T_H2$) cells in the absence of protective

antibodies[18,19,21–27]. Therefore, we tested the role of vaccine-induced antibodies and CD4[+] $T_H$ cells in iCoV2 + Alum-induced VAERD or iCoV2 + RIBI-mediated cross-protection. To test the role of antibodies in promoting viral clearance (iCoV2 + RIBI) or VAERD (iCoV2 + Alum), we challenged vaccine-naïve mice with SHC014 following passive serum transfer from vaccinated mice (Supplementary Fig. 10a). Consistent with our results with iCoV2 + RIBI vaccination (Fig. 2), serum from iCoV2 + RIBI-vaccinated animals promoted viral clearance compared to controls (Fig. 5a), suggesting that iCoV2 + RIBI induces cross-protective antibodies against SHC014. Somewhat surprisingly, serum transfer from iCoV2 + Alum-vaccinated mice also promoted viral clearance (Fig. 5a), indicating that iCoV2 + Alum-induced antibodies are not defective, nor do they cause antibody-dependent enhancement of infection. Further analysis found no effect of iCoV2 + Alum serum transfer on respiratory function (Fig. 5b). However, we did observe a modest but statistically significant increase in ALI and DAD without eosinophilic infiltration, in iCoV2 + Alum serum recipients (Fig. 5c–e). These results indicate that while iCoV2 + Alum vaccine-induced antibodies modestly contribute to pathology during heterologous infection, other immune system components drive iCoV2 + Alum-induced VAERD.

Given that iCoV2 + Alum induced type 2 inflammation during heterologous infection (Figs. 2f, g, 3), suggesting a strong $T_H2$-biased immune response, we next tested the role of $T_H$ cells via CD4[+] cell

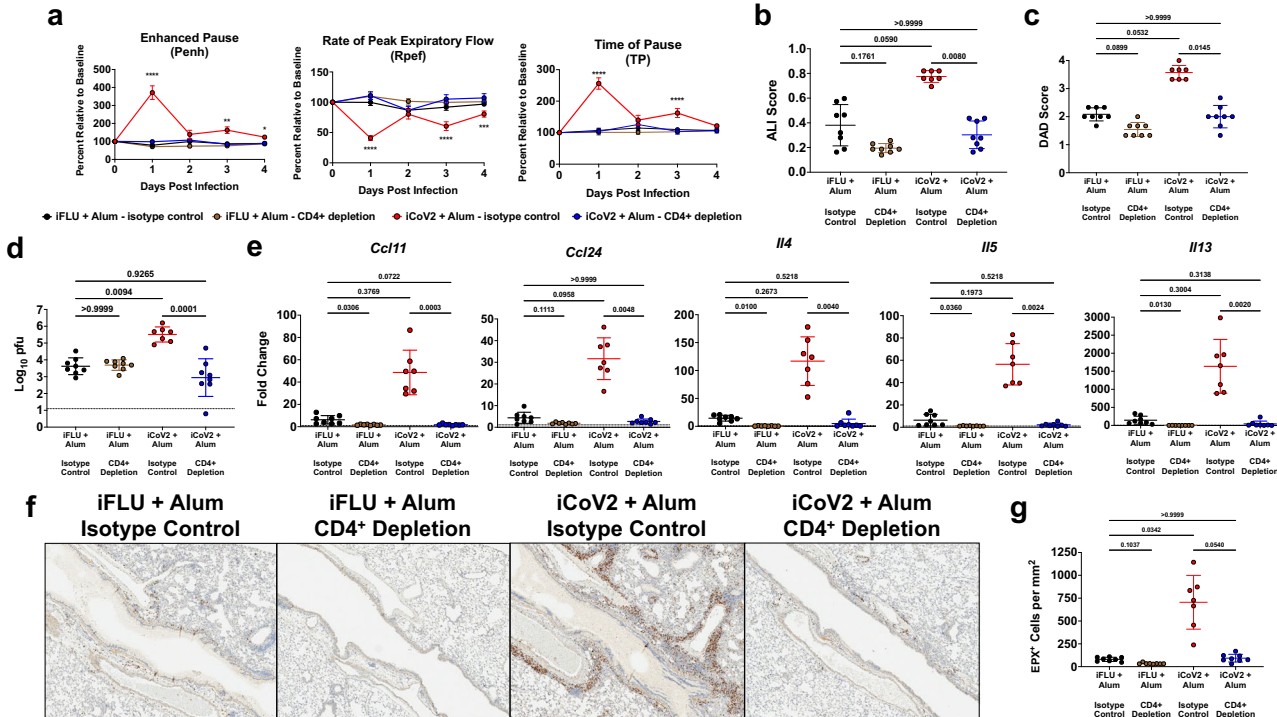

**Fig. 6 | CD4+ T helper cells promote vaccine-enhanced disease during heterologous infection. a** Pulmonary function measured by whole-body plethysmography following Rs-SHC014-CoV challenge in mice vaccinated with inactivated influenza virus adjuvanted with aluminum hydroxide (iFLU + Alum) or inactivated SARS-CoV-2 adjuvanted with aluminum hydroxide (iCoV2 + Alum) and administered anti-CD4 antibody (iFLU + Alum − CD4+ depletion, n = 12; iCoV2 + Alum − CD4+ depletion, n = 14) or isotype control antibody (iFLU + Alum − isotype control, n = 12; iCoV2 + Alum − isotype control, n = 13) prior to Rs-SHC014-CoV challenge; results reported as group mean ± standard error of the mean; ***p < 0.001, ****p < 0.0001, iCoV2 + Alum − isotype control versus iCoV2 + Alum − CD4+ depletion, two-way ANOVA with Geisser-Greenhouse correction and Dunnett's multiple comparisons correction. **b−e** Acute lung injury (ALI) scores (**b**), diffuse alveolar damage (DAD) scores (**c**), viral titers (pfu plaque forming units) (**d**), and type 2 cytokine gene expression (*Ccl11* C-C motif chemokine 11, *Ccl24* C-C motif chemokine 24, *Il4* interleukin 4, *Il5* interleukin 5, *Il13* interleukin 13) (**e**) in lungs at 5 DPI following Rs-SHC014-CoV infection in mice vaccinated with iFLU + Alum or iCoV2 + Alum and administered anti-CD4 antibody or isotype control antibody prior to Rs-SHC014-CoV challenge. **f, g** Representative photomicrographs (**f**) (scale bar = 0.5 mm) and quantification (**g**) of pulmonary eosinophils immunohistochemically labeled for eosinophil peroxidase (EPX, brown cells) at 5 DPI following Rs-SHC014-CoV infection in mice vaccinated with iFLU + Alum or iCoV2 + Alum and administered anti-CD4 antibody or isotype control antibody prior to Rs-SHC014-CoV challenge. **b−g** iFLU + Alum − isotype control (n = 8), iFLU + Alum − CD4+ depletion (n = 8), iCoV2 + Alum − isotype control (n = 7), iCoV2 + Alum − CD4+ depletion (n = 8). **b−e, g** Individual data points represent independent biological replicates; solid lines and error bars represent group mean ± standard deviation; data analyzed by Kruskal−Wallis test with Dunn's multiple comparisons correction; solid horizontal lines above data represent pairwise comparisons with p values. **d** Dotted line represents the assay limit of detection. Data presented as combined results from one (**b−g**) or two (**a**) independent animal experiments.

depletion in iCoV2 + Alum-vaccinated animals prior to SHC014 challenge (Supplementary Fig. 10b). Importantly, we observed that depletion of CD4+ cells reversed all measured signs of VAERD, including impaired respiratory function, pathology, delayed viral clearance, and type 2 inflammation (Fig. 6). Consistent with the hypothesis that a T$_H$2-biased response promotes VAERD, CD4+ cell depletion resulted in reversal of high type 2 cytokine expression (Fig. 6e) and significantly decreased pulmonary eosinophil infiltration (Fig. 6f, g). The striking impact of CD4+ cell depletion on multiple adverse outcomes strongly argues that T$_H$ cells are the major drivers of iCoV2 + Alum-induced VAERD.

## Discussion

Inactivated COVID-19 vaccines, of which approximately 5 billion doses have been administered, provide moderate protection against ancestral SARS-CoV-2 and early VOC[7]. However, these vaccines show significantly reduced efficacy against more recent VOC[8,12,17]. Given that other non-protective inactivated vaccines against RSV and measles virus have been associated with VAERD in both humans and animal models[18–27], we used established mouse models of homologous and heterologous viral challenge to test whether vaccine breakthrough with inactivated SARS-CoV-2 vaccines was associated with VAERD and whether this could be modulated by adjuvants. Our investigation

demonstrates that although iCoV2 + Alum protects against homologous and early pandemic VOC, the vaccine causes VAERD during heterologous infection by a SARS-r-CoV, SHC014, that is normally non-pathogenic in BALB/c mice. Importantly, this outcome can be avoided (and viral clearance accelerated) by using an alternative adjuvant (RIBI). Although three recent reports described enhanced pulmonary pathology in preclinical animal models using SARS-CoV-2 vaccines[42–44], here we report VAERD involving enhanced clinical disease, including impaired respiratory function, exacerbated pathology, and delayed viral clearance associated with a SARS-CoV-2 vaccine. Importantly, although eosinophils are a marker of type 2 inflammation and are often used as a marker of VAERD, we did not see a correlation between eosinophil numbers and other markers of disease, such as viral titers, CD4+ cell infiltration, ALI, or DAD, during infection at 10.5 months post-vaccination (Supplementary Fig. 7). This suggests that eosinophils may not play a pathogenic role in this model and that other disease markers should be considered in addition to eosinophils when assessing VAERD. Notably, in contrast to earlier challenge time points, we observed low eosinophil infiltration at this 10.5-month time point, which suggests that a lack of correlation between eosinophils and other parameters may depend on the time of challenge post-vaccination. Therefore, although other markers should also be considered, we believe that eosinophils should continue to be used as a

marker for the evaluation of VAERD. Finally, our findings highlight the possibility that some vaccinated individuals may be placed at increased risk of VAERD due to the emergence of a new coronavirus or the future evolution of more antigenically distinct SARS-CoV-2 variants.

Importantly, VAERD has not been reported in the context of COVID-19. Inactivated COVID-19 vaccines, which usually include Alum[7], have an established safety and efficacy record in humans, especially against early-pandemic strains[7,8,12]. However, as VOC like Omicron subvariants have become the dominant circulating strains, vaccine breakthrough has become a significant problem in individuals vaccinated with inactivated vaccines[8,12]. Importantly, these patterns were replicated in our models, in which iCoV2 + Alum protected against homologous challenge and an early-pandemic VOC but displayed breakthrough during heterologous challenge.

This raises the important question of whether the immunopathology observed in this mouse model reflects a risk for humans. Inactivated vaccines induce protective immunity against homologous viruses in both mouse and non-human primate studies, consistent with the performance of these vaccines during the early stages of the pandemic[7,10,11,13–16,42]. Our results suggest that adverse outcomes would not have been observed during the early stages of the pandemic, when inactivated vaccines provided substantial protection against circulating strains. There are also no reports of enhanced pathology or VAERD following exposure to more antigenically distinct Omicron variants. Furthermore, while we observed type 2 inflammation following challenge with either ancestral SARS-CoV-2 or an early-pandemic VOC (B.1.351), iCoV2 was protective against both of these viruses, and the more severe disease manifestations were only apparent following challenge by SHC014, which represents a potential emerging disease threat. This suggests that exposure to new heterologous coronaviruses poses the highest risk of VAERD. However, we believe that our findings support further investigation, both in animal challenge models and through surveillance of vaccinated human populations, of potential VAERD risks associated with newly arising antigenically distinct SARS-CoV-2 variants.

While the direct relevance of these findings to humans remains to be determined, we believe that this work illustrates the importance of rigorously testing vaccine performance against heterologous viruses. Liu et al. previously evaluated an Alum-adjuvanted inactivated SARS-CoV-2 vaccine for efficacy against both SHC014 and another SARS-r-CoV, WIV1-CoV, and showed partial protection against SHC014 challenge in a human ACE2 transgenic mouse model[57]. While the differences between our work and that of Liu et al. may reflect differences in the inactivation methods used for the two vaccines, or mouse-strain specific differences, it is also important to highlight that the Liu studies did not evaluate the respiratory function, ALI, DAD, or pulmonary type 2 inflammation in infected animals. Therefore, our results, along with our prior work with SARS-CoV-1 vaccines[29], emphasize the importance of comprehensive analysis of coronavirus vaccines, as performed in this study, to identify adverse outcomes associated with vaccine breakthrough.

The disease signs and pulmonary pathology observed in iCoV2 + Alum-vaccinated mice during SHC014 infection are similar to phenotypes observed in models of RSV VAERD, including increased pulmonary eosinophil infiltration, inflammatory damage, and impaired respiratory function[18–27]. These prior studies with RSV demonstrated that the inactivation method used to generate the vaccine destroyed protective epitopes, causing vaccine breakthrough and VAERD driven by immune responses against non-protective epitopes. The inactivated SARS-CoV-2 vaccine used in this study elicited protective immunity against mouse-adapted viruses based on ancestral SARS-CoV-2 (SARS-CoV-2-MA10) and the B.1.351 VOC (MA10-B.1.351), while iCoV2 + RIBI promoted clearance of SHC014, which argues against a loss of protective epitopes. Instead, we believe that the antigenic

mismatch between the vaccine and SHC014, in combination with the type 2-biased immune response associated with Alum, is the primary cause of vaccine failure and VAERD. While passive transfer of serum from iCoV2 + Alum-vaccinated mice induced modest immunopathology upon SHC014 challenge, CD4+ cell depletion completely ameliorated the delayed viral clearance and pulmonary pathology in SHC014-infected mice. This suggests that $T_H$ cell epitopes conserved between SARS-CoV-2 and SHC014 are the major determinants of VAERD. Therefore, to assess whether VAERD is unique to inactivated SARS-CoV-2 vaccines or is a potential outcome with other vaccine platforms, it will be important to determine whether these adverse responses are driven by epitopes within the spike protein, which is present in all SARS-CoV-2 vaccines, or by epitopes within the highly conserved nucleocapsid, envelope, and membrane proteins present in whole virus-based inactivated vaccines[7].

As discussed above, our observations are reminiscent of VAERD caused by the failed formalin-inactivated RSV vaccine tested in the 1960s, which caused exuberant type 2 inflammation and eosinophilic and neutrophilic pulmonary infiltrates in vaccinated subjects[18–27]. Given the type 2 cytokine response observed in iCoV2 + Alum-vaccinated animals and the fact that $T_H$ cells are critical to the adverse vaccine responses, we hypothesize that $T_H2$ cells infiltrate infected pulmonary tissue and drive type 2 inflammation via secretion of cytokines like interleukin (IL)-4, IL-5, and IL-13. This suggests that anti-atopic therapies targeting these cytokines may be beneficial should VAERD be observed in humans.

Our results also raise the question of whether it is possible to reprogram type 2-biased responses elicited by iCoV2 + Alum and thereby reduce the risk of VAERD. Despite potential biological limitations, the practicality of this approach is supported by its similarity to current real-world vaccination strategies[8,58–63]. Our observation that follow-up boost vaccination with S2P + RIBI partially reduced VAERD in mice initially vaccinated with iCoV2 + Alum suggests that this may be a promising strategy to reduce or prevent VAERD. Future studies will be required to elucidate the protective mechanism(s) underlying the S2P + RIBI alternative boost. Due to high levels of background serum neutralization activity with SHC014, which was worse in the older mice used in this study, we were unable to detect S2P + RIBI-induced cross-neutralizing titers against SHC014 (Supplementary Fig. 9). Based on these results, we cannot determine whether S2P + RIBI induced low levels of cross-neutralizing antibodies that were not detected by our assay and/or this vaccine promoted a degree of cross-protection via other adaptive immune components, including non-neutralizing antibodies and/or optimized cellular responses. Lastly, it will be important to determine whether infection with SARS-CoV-2 resets inactivated COVID-19 vaccine + Alum-induced type 2 immunity, as this would have important implications for the long-term risk for inactivated vaccine recipients should a novel heterologous SARS-r-CoV emerge in a vaccinated population.

The BALB/c model used in this study is predisposed to strong type 2-biased immune responses[64,65], which likely increases susceptibility to type 2 VAERD. This has important implications for individuals vaccinated with Alum-adjuvanted inactivated vaccines who may be at risk of adverse outcomes, particularly individuals predisposed to atopic immune responses. However, prior work with coronavirus vaccines has demonstrated that vaccination can induce enhanced pulmonary eosinophil infiltration following viral challenge in both BALB/c and C57BL/6 mice, as well as ferrets and NHPs, which suggests that adverse effects occur across a range of genetic backgrounds and species[4,5,21,28–41]. However, it is also important to note that while multiple models have noted enhanced type 2 inflammation, we also identified more severe disease manifestations in our model with heterologous challenge. Therefore, it will be important to extend this work to other model systems, including additional mouse strains or species, while also carefully assessing inactivated vaccine responses in

humans. It is also important to consider that our studies were performed in young, immunologically naïve mice. Therefore, to more effectively model vaccine effects in humans, it will be important to consider factors such as host genetic variation, prior coronavirus exposure, advanced age, or underlying medical conditions that might affect inactivated vaccine performance and impact susceptibility to VAERD.

This study demonstrates that adjuvant choice is critical to optimizing vaccine design. Our results indicate that Alum, which is often reported to promote type 2 immune responses in preclinical models[19,66–68], is a major determinant of VAERD in this model and suggests that inactivated vaccines formulated with Alum may exhibit a suboptimal safety profile in the context of heterologous infections. In contrast, our finding that using RIBI averts VAERD and even promotes more efficient viral clearance has potentially important implications for universal coronavirus vaccine design, including adjuvant selection. These results are consistent with the findings of Iwata-Yoshikawa and Shiwa et al., who found that the use of an alternative adjuvant consisting of a combination of toll-like receptor agonists (which the authors described as a $T_H1$-$T_H2$ balanced adjuvant), instead of Alum or no adjuvant, promoted improved clinical protection and avoided type 2 immunopathology in a SARS-CoV-2 mouse model of VAERD[44]. Together with our findings, the study by Iwata-Yoshikawa and Shiwa et al. illustrates that adjuvants have profound impacts on the nature of vaccine-induced immune responses, including inflammatory cellular and cytokine profiles and the quality of the humoral response, and can thus determine the outcome of viral infection in vaccinated individuals.

Although it remains unclear what immune system components induced by iCoV2 + RIBI vaccination are responsible for heterologous protection, we found that passive serum transfer from vaccinated mice to naïve recipients resulted in significantly improved clearance of SHC014. We also observed a modest induction of SHC014-specific neutralizing antibody by iCOV2 + RIBI vaccination. However, we and others have also demonstrated that non-neutralizing antibodies can mediate cross-protection during coronavirus infection[69,70]. Therefore, studies are underway to resolve the relative contribution of neutralizing and non-neutralizing antibodies, as well as cellular immunity against conserved epitopes, in mediating cross-protection.

In summary, our findings highlight the potential impact of different adjuvants on the safety and efficacy of inactivated SARS-CoV-2 vaccines in the context of infections by both SARS-CoV-2 VOC and highly heterologous SARS-r-CoVs. Unlike the COVID-19 pandemic, possible future coronavirus epidemics will occur in the context of widespread preexisting SARS-CoV-2 immunity acquired through infection and/or vaccination. This critical new variable is almost certain to impact the course of future SARS-r-CoV epidemics and should therefore be incorporated into pandemic preparedness strategies. This model anticipates continued SARS-CoV-2 variant evolution and future SARS-r-CoV emergence as important considerations in assessing vaccine safety and efficacy and represents a reproducible approach to predictively model such variables. By elucidating factors that drive beneficial or harmful cross-reactive vaccine-induced immune responses during heterologous infection, this investigation advances the development of safe and effective vaccination strategies that will increase SARS-r-CoV pandemic preparedness while mitigating potential adverse outcomes like VAERD. These findings are also useful for the development of pan-coronavirus vaccines and inform the potential utility of authorized SARS-CoV-2 vaccines during the early stages of future SARS-r-CoV epidemics.

## Methods
### Viruses
Infectious clone (ic)SARS-CoV-2 wild-type (derived from the D614 strain based on the WA1 sequence)[51], mouse-adapted SARS-CoV-2-MA10[49,50], mouse-adapted MA10 expressing SARS-CoV-2 B.1.351 spike[56] and SHC014[4], as well as reporter viruses icSARS-CoV-2-nLuc[49,50], icSARS-CoV-2-B.1.351-nLuc[56] and icSHC014-CoV-nLuc[56,71] were cultured on Vero E6 (USAMRIID) cells in Dulbecco's Modified Eagle Medium (DMEM, Gibco) containing 5% heat-inactivated FBS (HI-FBS). The icSARS-CoV-2 wild-type strain and the mouse-adapted SARS-CoV-2-MA10 are available through BEI Resources. Other viruses used, including reporter viruses, may be available from the laboratory of Ralph Baric upon request. The SHC014 virus used in this study is not genetically modified (i.e., SHC014 used here is a wild-type full-length infectious clone and is not mouse-adapted or chimeric). All activities involving coronaviruses were performed in an approved and registered biosafety level 3 (BSL3/ABSL-3) facility following Standard Operating Procedures by trained personnel wearing appropriate personal protective equipment, including Powered Air Purifying Respirators, in accordance with the guidelines outlined in the CDC/NIH *Biosafety in Microbiological and Biomedical Laboratories* (6th edition), as well as the *NIH Guidelines for Research Involving Recombinant or Synthetic Nucleic Acid Molecules* (April 2016). Additionally, Tecniplast Sealsafe HEPA-filtered rodent housing was used for experiments involving viral infection of mice.

### Vaccines
iCoV2 was produced as previously published using icSARS-CoV-2, which was derived from the D614 strain based on the WA1 sequence[51], following the method of ref. 72. Culture supernatants from Vero E6 (USAMRIID) cells (kindly provided by Ralph Baric) seeded with wild-type SARS-CoV-2 were collected and centrifuged to remove cell debris. The resulting clarified supernatant was treated with 0.05% formalin for 48 h at 4 °C. The formalin-inactivated virus was exposed to 25 mJ UV light, then placed in a polyallomer ultracentrifuge tube underlaid with 20% sucrose and centrifuged at 78,000×g overnight. The pellet containing the purified inactivated virus was recovered in PBS and frozen at −80 °C until use. For each vaccination, the vaccine was mixed with adjuvant per manufacturer protocols resulting in the delivery of 0.2 µg of adjuvanted vaccine in 10 µL volume. Inactivated A/PR8 influenza virus (Charles River Laboratory) was also prepared with adjuvant per manufacturer protocols with a final inactivated vaccine dose of 0.2 µg in 10 µL volume. Vaccines were delivered to the left rear footpad. Adjuvants used were Alum (Alhydrogel, Invivogen) and RIBI (Sigma Adjuvant System, Sigma Aldrich).

### Mouse vaccination and challenge model
All mouse studies were conducted under protocols (18-300 and 21-259) approved by the Institutional Animal Care and Use Committee (IACUC) at the University of North Carolina at Chapel Hill, an AAALAC International-accredited institution, in alignment with the recommendations outlined in the *Guide for the Care and Use of Laboratory Animals* (8th edition). BALB/cAnNHsd mice were purchased from Envigo/Inotiv (Stock 047) and housed in our ABSL-3 facility on a 12:12 light cycle, kept within a temperature range of 20–23.3 °C and humidity range between 30–70%. Autoclaved cages (Tecniplast, EM500) were used with irradiated Bed-o-Cob (ScottPharma, Bed-o-Cob 4RB), ad libitum-irradiated chow (LabDiet, PicoLab Select Rodent 50 IF/6 F 5V5R) and autoclaved water bottles. Cages were changed at least every 14 days and water bottles were changed every 7 days. Only female mice were used due to the difficulty of maintaining male health related to aggressive behavior in a co-housing context; given the length of experiments and sample size requirements, use of female mice was optimal.

Young adult (6–8 weeks old) female BALB/cAnNHsd mice were lightly anesthetized with isoflurane and vaccinated with 0.2 µg of iCoV2 or iFLU (used for mock-vaccination) delivered in a 10 µL volume into the left rear footpad. In selected experiments (Supplementary Figs. 1, 3), phosphate-buffered saline (PBS) alone was used for mock-

vaccination. Mice were boosted with 0.2 µg of vaccine three weeks post-initial vaccination. Submandibular bleeds were collected pre-prime, 3 weeks post-prime, and 3 weeks post-boost. Approximately 3 weeks post-boost, vaccinated and boosted mice were lightly anesthetized with 50 mg per kg ketamine + 5 mg per kg xylazine and challenged intranasally with 50 µL of MA10 ($1 \times 10^4$ plaque-forming units [pfu]), MA10-B.1.351 ($5 \times 10^4$ pfu) or Rs-SHC014-CoV (SHC014) ($1 \times 10^5$ pfu), or mock challenged with 50 µL of PBS alone. Post-challenge, mice were monitored, weighed, and scored for clinical signs daily. Mice were euthanized at 2 or 5 DPI unless a mouse reached humane endpoint criteria (clinical score of 4 or higher) before 5 DPI. The clinical scoring scheme is as follows: 0 = clinically normal; 1 = piloerection; 2 = piloerection and kyphosis; 3 = piloerection, kyphosis, and reduced movement; 4 = markedly reduced movement and/or dyspnea; 5 = moribund, dead, or euthanized[73]. Mice were euthanized by an overdose of isoflurane anesthesia (Baxter), blood was collected by cardiocentesis, and tissues were collected for post-mortem analysis.

## Multiplex ACE2 inhibition assay

SARS-CoV-2 spike, along with circulating variants, such as Alpha, Beta, Delta, and Omicron subvariants in the multiplexed Meso Scale Discovery (MSD) V-PLEX SARS-CoV-2 Panel-25, were used to measure ACE2 blocking antibodies. Briefly, 96-well plates were blocked using MSD Blocker A for 30 min and washed. Vaccinated and mock-vaccinated control serum samples (diluted 1:50) and calibrator standards were added to the plate and incubated for 1 h at 22 °C, shaking at 700 rpm. After incubation, MSD SULFO-tagged Human ACE2 was added to the wells for detection, incubated at 22 °C for 1 h, and then washed. The plate was read on the MESO QuickPlex SQ 120 instrument, and ACE2 blocking activity was analyzed using the equation: ([1 − Average Sample ECL Signal ÷ Average ECL signal of the blank well] × 100).

## Pathology

Left lung lobes of mice were collected at necropsy, infiltrated with 100 µL 10% neutral buffered formalin (NBF), and then immersion-fixed in 10% NBF for 7 days. After the transfer of the fixed lung lobes to a new tube of 10% formalin, the lungs were removed from the BSL3 facility for preparation for histology submission. Lungs were rinsed with PBS (Gibco), placed in cassettes, and stored in 70% ethanol until the tissue was paraffin-embedded and sectioned. Specimens were processed on an automated tissue processor (Leica ASP 6025), embedded in paraffin (Leica Paraplast), sectioned at 5-µm thickness, and stained with hematoxylin and eosin (H&E, Richard Allan Scientific). For embedding, lungs were placed in a standardized orientation to best visualize the main bronchus.

Lung histopathology was evaluated and scored by an American Board of Veterinary Practitioners (ABVP)-certified veterinary pathologist on H&E-stained sections. Lung pathology was quantified using two scoring systems previously validated for respiratory coronavirus infection in mice[49,50,74], with the pathologist blinded to the status of the study groups. Three representative alveolar high power fields (400X total magnification) were selected per H&E tissue section and scored using previously published semi-quantitative ALI and DAD scoring systems[52–55]. Briefly, ALI scores were determined as follows: (A) polymorphonuclear leukocytes in alveolar spaces (none = 0, 1–5 cells = 1, > 5 cells = 2); (B) polymorphonuclear leukocytes in alveolar septae (none = 0, 1–5 cells = 1, > 5 cells = 2); (C) well-formed hyaline membranes (none = 0, one membrane = 1; > 1 membrane = 2); (D) proteinaceous material/debris in air spaces (none = 0, one area = 1, > 1 area = 2); (E) alveolar septal thickening (> 2x mock animal thickness = 0, 2–4x mock thickness = 1, > 4x mock thickness = 2). ALI scores were calculated as follows: ([20 x A] + [14 x B] + [7 x C] + [7 x D] + [2 x E]) ÷ 100. DAD scores were determined as follows: 1 = within normal limits; absence of cellular degeneration, sloughing, and necrosis;

2 = uncommon solitary cell sloughing and necrosis, ≤ 3 foci per HPF; 3 = multifocal (> 3 foci per HPF) cellular degeneration, sloughing, and necrosis ± septal wall hyalinization/early hyaline membrane formation; 4 = severe (> 75% of the field) cellular degeneration and sloughing with prominent necrosis, or the presence of at least one well-formed hyaline membrane. For each ALI and DAD score, the average of scores from three representative fields per tissue section determined the final score for the specimen.

## Immunohistochemistry

Formalin-fixed tissues were processed on a Leica ASP 6025 tissue processer, embedded in paraffin (Leica Paraplast), and sectioned at 5-µm thickness onto positively charged slides. Sequential tissue sections were labeled for antigens using anti-CD4 monoclonal antibody (1:1000 dilution; ab183685, Abcam), anti-C3 Complement antiserum (1:100 dilution; 55444, MP Biomedical), or anti-Arginase-1 monoclonal antibody (1:100 dilution; 93668S, Cell Signaling Technology) on the Ventana Discovery automated staining platform (Roche), or with anti-SARS nucleocapsid polyclonal antibody (1:8000 dilution; NB100-56576, Novus Biological) or anti-EPX polyclonal antibody (1:1000 dilution; PA5-62200, Invitrogen) on the Bond III (Leica Biosystems) automated stainer. Briefly, for labeling performed on the Ventana Discovery platform, antigen retrieval was accomplished using CC1 pH 8.5 (950-500, Roche) or Protease 2 (760-2019, Roche). After pretreatment, tissues were blocked, and then incubated with either an anti-CD4 monoclonal antibody at 1:1000, an anti-C3 Complement antiserum at 1:100, and an anti-Arginase-1 monoclonal antibody at 1:100 for 1 h. Ready-to-use (RTU) secondary antibodies used were Discovery OmniMap polyclonal anti-rabbit horseradish peroxidase (HRP) (RTU; 760-4311, Roche) or polyclonal anti-goat HRP (RTU; Dako, P0160), followed by stain development with Discovery Chromo Maps 3,3′-diaminobenzidine (DAB) (760-159, Roche) and Hematoxylin II (790-2208, Roche) for nuclear staining. For labeling performed on the Bond platform, slides were dewaxed in Bond Dewax solution (AR9222) and hydrated in Bond Wash solution (AR9590, Leica). Heat-induced antigen retrieval was performed at 100 °C in either Bond-Epitope Retrieval solution 1 pH-6.0 (AR9961, Leica) or Bond-Epitope Retrieval solution 2 pH-9.0 (AR9640, Leica). After pretreatment, tissues were blocked, and then incubated with either anti-EPX polyclonal antibody at 1:1000 or anti-SARS nucleocapsid polyclonal antibody at 1:8000 dilution for 1 h, followed by Novolink Polymer polyclonal anti-rabbit HRP secondary antibody (RTU; RE7260-K, Leica). Antibody detection with DAB was performed using the Bond Intense R detection system (DS9263, Leica). Stained slides were dehydrated and coverslipped with Cytoseal 60 (23-244256, Thermo Fisher Scientific). A positive control was included for each assay.

For image analysis, a composite image composed of 5 × 5 fields at 200X magnification of the inferior section of a single section of the left lung was collected (3.53–3.57 mm² total area). Image analysis was performed using Nikon Elements software. A threshold for positive staining was set, and signal intensity was categorized as low-, medium-, or high-intensity. Positivity and signal intensity thresholds were set using positive and negative control samples. To increase the specificity of detection for the desired cell types ($EPX^+$ eosinophils or $CD4^+$ $T_H$ cells), low-intensity signals were excluded from the analysis. Exclusion criteria were pre-established and were applied equally to all groups and samples. Thresholds for area, circularity, and equal diameter were set for positive object count. Results were reported as positive cell density (number of $EPX^+$ or $CD4^+$ cells per mm²).

## Alcian blue−periodic acid-Schiff (AB/PAS)

AB/PAS stains were used to identify mucus in pulmonary airways. Samples were first baked at 60 °C for 60 min minimum, deparaffinized in xylene, and hydrated with graded ethanol. Tissue sections were then stained with AB/PAS using the Leica Autostainer XL Staining System

ST5010 (Leica). The slides were stained with Alcian Blue (867, Anatech, LTD) for 10 minutes, immersed in Periodic Acid (A223-100, Thermo Fisher Scientific) for 5 min, rinsed in water, then transferred to Schiff reagent (SS32-500, Fisher Scientific) for 30 min followed by a Sulfurous rinse for 1 min, and finally washed in running tap water for 10 min. After staining, slides were then dehydrated and coverslipped with Cytoseal 60 (23-244256, Thermo Fisher Scientific). A positive control slide was included for each run of this assay.

## Viral titers

At necropsy, the superior and middle lung lobes were collected and frozen in a tube containing DMEM (Gibco) with 5% heat-inactivated FBS with glass beads and frozen at −80 °C. At the time of assay, tissues were thawed, homogenized for 40 s at 6000 rpm using a MagNA Lyser (Roche), and centrifuged to clarify the sample from residual tissue and beads. A 50 µL aliquot of clarified homogenate was removed and added to 450 µL of DMEM + 5% HI-FBS. The resulting dilution was used to make additional tenfold dilutions. An aliquot (200 µL) of each dilution was plated in duplicate in 12-well plates containing Vero E6 (USAMRIID) monolayers. Plates were gently rocked every 15 min to ensure uniform distribution of virus across the monolayer. After 1 h of rocking, carboxymethylcellulose (CMC) overlay (1.25% CMC and 1X Alpha Minimum Essential Medium) was added to each, and the plates were incubated at 37 °C for 4 days, at which point plates were fixed with 4% paraformaldehyde overnight. After fixation, fix/overlay was removed, cell monolayers were stained with 0.25% crystal violet, and plaques were counted.

## Quantitative RT-PCR

At necropsy, postcaval lung lobes were collected for RNA isolation. Lung lobes were placed in TRIzol (Invitrogen) with glass beads and homogenized at 6000 rpm for 40 s using a MagNA Lyser (Roche) homogenizer. The homogenates were clarified by centrifugation at 10,640×g for 2.5 min. Clarified samples were transferred to a second tube and removed from the BSL3 facility. RNA was isolated from the sample using the Qiagen RNeasy Kit (Qiagen) per the manufacturer's protocol. Using the High-Capacity cDNA reverse transcription kit (Applied Biosystems), cDNA was synthesized by adding 1 µg of RNA per reverse transcription reaction. Quantitative PCR was performed for each of the primer-probe sets using the TaqMan Fast Advanced Master Mix (Applied Biosystems). Two µL of cDNA were added per reaction. All genes were normalized to *Gapdh* expression and reported as fold change using the ΔΔCt method. The limit of detection was determined by water controls and set for all primers at a Ct of 34.

Primer-probe sets used:

*Gapdh* (Applied Biosystems, Catalog # 4352339E, Probe VIC/MGB);

*Ccl11* (Integrated DNA Technologies, Catalog # Mm.PT.58.28587819, Probe FAM/ZEN/IBFQ);

*Ccl24* (Integrated DNA Technologies, Catalog # Mm.PT.58.13396581, Probe FAM/ZEN/IBFQ);

*Il4* (Integrated DNA Technologies, Catalog # Mm.PT.58.7882098, Probe FAM/ZEN/IBFQ);

*Il5* (Integrated DNA Technologies, Catalog # Mm.PT.58.41498972, Probe FAM/ZEN/IBFQ);

*Il13* (Integrated DNA Technologies, Catalog # Mm.PT.58.31366752, Probe FAM/ZEN/IBFQ).

Additional information related to primer-probe sets is provided in Supplementary Data 1.

## Neutralization assay

Post-boost serum samples from vaccinated mice were evaluated for neutralizing antibody levels using an established SARS-CoV-2 neutralization assay[51,56,71]. Blood was collected from vaccinated mice 19–21 days post-boost vaccination and centrifuged at 5000×g for 5 min. Immune serum samples were heat-inactivated at 56 °C for 30 min and centrifuged at 10,000×g for 15 min. Threefold serial dilutions of individual serum samples were mixed with equal amounts of diluted icSARS-CoV-2-nLuc (D614G)[51], icSARS-CoV-2-B.1.351-nLuc[56], or icSHC014-CoV-nLuc[56,71] and incubated at 37 °C with 5% $CO_2$ for 1 h. Following incubation, virus-serum mixtures were added to duplicate wells in a 96-well dish containing Vero E6 C1008 cells (ATCC) and incubated for 24 h. Virus-only controls as well as cell-only controls were included in each neutralization assay plate. Luciferase activity was then measured via the Nano-Glo Luciferase Assay System (Promega) according to the manufacturer's protocol. Neutralization titers were defined as the sample dilution at which an 80% reduction in relative light units was observed relative to the average of the control wells (80% reciprocal inhibitory concentration [$IC_{80}$]). For the icSHC014-CoV-nLuc assay, normal mouse serum shows high background neutralizing activity against SHC014, which reduced the sensitivity of the assay.

## Whole-body plethysmography (WBP)

WBP measurement was performed using Buxco Small Animal WBP (Data Sciences International [DSI]) and analyzed using Buxco Fine-pointe Software (DSI) as previously described[75]. Briefly, measurements were acquired for individual mice using the COPD study type and WBP Volume apparatus at baseline and once each day, 1 through 4 DPI (i.e., measurements included: Baseline, 1 DPI, 2 DPI, 3 DPI, and 4 DPI). Reported measurements are the average parameter value measured during a 10-min data acquisition period following a 20-min acclimation period. One through four DPI measurements were reported as percentage relative to baseline measurement.

## Passive serum transfer

Donor BALB/c mice were vaccinated using the prime-boost method with either iFLU + Alum, iCoV2 + Alum, or iCoV2 + RIBI (Mouse vaccination and challenge model, Methods). At 4 weeks post-boost vaccination, blood was collected from sacrificed donor mice via cardiac puncture. Individual blood samples were incubated for at least 30 min to allow clotting and centrifuged at 5000×g for 5 min for serum separation. Equal volumes of individual serum samples were pooled within each vaccine group. Approximately 225 µL of fresh pooled immune serum was transferred to age-matched naïve recipient mice via intraperitoneal injection 1 day prior to the challenge with SHC014.

## CD4⁺ cell depletion

BALB/c mice were vaccinated using a double-boost method with either iFLU + Alum or iCoV2 + Alum. To increase the power of individual experiments, after vaccination using the prime-boost method (Mouse vaccination and challenge model, Methods), mice were subsequently administered a third dose of the same vaccine (see Fig. 4; iCoV2 + Alum double-boost group exhibits higher magnitude and lower variability in some disease parameters). At 4 weeks post-second boost vaccination, anti-CD4 monoclonal antibody (GK1.5; BE0003-1, BioXCell) or isotype control monoclonal antibody (LTF-2; BE0090, BioXCell) were administered to mice via intraperitoneal injection in 250 µL of PBS at day -5 (500 µg per mouse), day -3 (250 µg per mouse), and day 2 (125 µg per mouse) relative to challenge at day 0 with SHC014.

## RNA sequencing (RNA-Seq)

RNA was extracted as described (Quantitative RT-PCR, Methods). Quality (RIN and DV200) was assessed via Tapestation (Agilent). Samples with a RIN < 2 were excluded from moving forward. Libraries for RNA-Seq were generated at the UNC High Throughput Sequencing Facility with a Kapa total RNA stranded library prep with Ribo Erase. Samples were barcoded and pooled before running on an Illumina NovaSeq (whole S4 flow cell). We generated 1 × 50 SE reads at a median coverage of 115.47 million reads (Range: 69.25–185.95 million reads). We ran FastQC (v0.11.9)[76] to confirm data quality, with all samples passing. Subsequently, we ran Salmon (v1.10.0)[77] to quantify

transcripts. We used alignment based mode, adjusting for GC bias in reads, and bootstrapping with inferential replicates. Following quantification, we imported gene-level count matrices using *tximport* (v1.28.0)[78] for use in differential expression analyses. We normalized count data with DESeq1 (v1.40.2)[79], and generated logarithmic fold changes and metrics of significance (raw and Bonferroni-corrected $p$ values, false discovery rate q-statistics). DEG with adjusted $p$ values of < 0.05 were run through GO analyses via PANTHER 17.0[80–82]. All analyses were run in Bioconductor and the R statistical package.

### Statistics

Statistical analyses were performed using GraphPad Prism 9 and 10 (except for RNA-Seq analysis). Evaluation for differences between group means were evaluated using (i) Kruskal–Wallis with Dunn's multiple comparisons correction or (ii) ordinary two-way analysis of variance (ANOVA) with Tukey's multiple comparisons correction (for grouped analyses). Multivariable correlation analysis (Supplementary Fig. 7) was performed to compute nonparametric Spearman $r$ values. Specific comparisons evaluated for each data set are specified in the figures. Statistical tests are corrected for multiple comparisons and/or repeated measures. For all statistical analyses, pairwise multiplicity-adjusted $p$ values (two-tailed) were calculated. When appropriate, data were presented as scatter dot plots showing individual data points representing independent biological replicates (discrete samples from individual animals) with group mean values represented using a horizontal line. Body weights, clinical scores, and pulmonary function (WBP) were analyzed via repeated measurements of the same individual animals. Two-sided error bars represent the standard deviation or standard error of the mean (specified in figure legends). When possible, $p$ values are represented numerically in data figures showing pairwise comparisons. For body weights and WBP, asterisks (* or #) are used to denote $p$ values using the following scheme: (* or #) = 0.01–0.05, (** or ##) = 0.001–0.01, (*** or ###) 0.0001–0.001, (**** or ####) < 0.0001.

### Reporting summary

Further information on research design is available in the Nature Portfolio Reporting Summary linked to this article.

## Data availability

Source data are provided as a Source Data file. RNA Sequencing raw .fastq data files are submitted to the Sequence Read Archive (SRA) database under BioProject ID PRJNA1022427, accession codes SRX21928707 through SRX21928826 [https://www.ncbi.nlm.nih.gov/sra/?term=PRJNA1022427]. Source data are provided with this paper.

## Code availability

Custom computer code or mathematical algorithms were not used to collect or analyze data. RNA sequencing data were analyzed using publicly available software (RNA sequencing [RNA-Seq], Methods).

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

## Acknowledgements

This project was supported by grants from the National Institutes of Health: 1P01AI158571, U19 AI100625, R01 AI157253, U19 AI109680, K01 OD026529, 5T32 AI007419-30, T32 GM008719, P30CA016086, U01AI149644, and U54 CA260543. This work was also generously supported by institutes and awards affiliated with the University of North Carolina School of Medicine: Infectious Disease Drug Discovery Program (ID3), North Carolina Translational Clinical Sciences Institute (NCTraCS) and Emerging Challenges in Biomedical Research COVID Pilot Award, the Junior Investigator Development Award, and by the North Carolina State Policy Collaboratory at the University of North Carolina at Chapel Hill. This project was also supported by the Rapidly Emerging Antiviral Drug Development Initiative at the University of North Carolina at Chapel Hill with funding from the North Carolina Coronavirus State and Local Fiscal Recovery Funds program, appropriated by the North Carolina General Assembly. We thank the following individuals for providing thoughtful guidance, critical analysis, and other support that contributed to the completion of this project: Melissa Mattocks, Ande West, Bentley Midkiff, Dr. Jonathan Schisler, Dr. Timothy Sheahan, Dr. Jack Harkema, Dr. Richard Boucher, and Dr. Barton Haynes.

## Author contributions

Designed study and formulated hypotheses: J.A.D., S.A.T.-B, M.T.F., V.K.B. and M.T.H. Designed experiments: J.A.D., S.A.T.-B., M.T.F., V.K.B., and M.T.H. Conducted experiments: J.A.D., S.A.T.-B., A.C.K., E.J.A., K.D.P., B.P., S.A. Martinez, J.L.D., S.S., E.A.M., G.D., L.E.A., I.N.C., N.L.M., J.B., M.K.D.E., L.M.R., N.C.P., V.K.B. Acquired data: J.A.D., S.A.T.-B., A.C.K., E.J.A., K.D.P., S.A. Martinez, S.S., and I.N.C. Analyzed data: J.A.D., S.A.T.-B, A.C.K., S.A. Martinez, S.S., I.N.C., C.L.L., S.A. Montgomery, P.L., M.T.F., V.K.B. and M.T.H. Interpreted data: J.A.D., S.A.T.-B., A.C.K., S.S., I.N.C., C.L.L., P.L., R.S.B., M.T.F., V.K.B. and M.T.H. Provided scientific expertise, technical services, and reagents: J.A.D., S.A.T.-B., A.C.K., E.J.A., G.D., L.E.A., K.H.D. III, S.R.L., D.R.M., A.S., J.M.P., B.L.Y., M.K.D.E., L.M.R., N.C.P., C.L.L., S.A. Montgomery, P.L., R.S.B., M.T.F., V.K.B. and M.T.H. Wrote manuscript: J.A.D., S.A.T.-B., A.C.K., G.D., I.N.C., C.L.L., S.A. Montgomery, P.L., R.S.B., M.T.F., V.K.B., and M.T.H. Edited manuscript: J.A.D., S.A.T.-B., A.C.K., S.A. Martinez, G.D., K.H.D. III, L.E.A., S.R.L., J.M.P., C.L.L., R.S.B., M.T.F., V.K.B. and M.T.H. All authors read the final version of the manuscript and approved the submission.

## Competing interests

R.S.B. has served on the Scientific Advisory Boards for Takeda vaccines, VaxArt and Invivyd Therapeutics, and has collaborations with Gilead, Janssen Pharmaceuticals, Pardas Biosciences, and Chimerix. M.T.H. has collaborations with Moderna, Inc. and Chimerix. R.S.B., K.H.D. III, and S.R.L. are listed as inventors on patents pertaining to the mouse-adapted SARS-CoV-2 viruses (MA10 and MA10-B.1.351; Patent number 11,225,508) and the SARS-CoV-2 nanoLuciferase viruses (SARS-CoV-2-nLuc and B.1.351-nLuc; Patent number 11,492,379) used in this study. The remaining authors declare no competing interests.
