## [Peer Review File · Nature Communications]

Adjuvant-dependent impact of inactivated SARS-CoV-2 vaccines during heterologous infection by a SARS-related coronavirusReviewers' Comments:

Reviewer #1:

Remarks to the Author:

Inactivated SARS-CoV-2 vaccines adjuvanted with Alum have been administered to millions of people worldwide. In this manuscript Dillard et al. demonstrate that inactivated SARS-CoV-2 vaccines adjuvanted with Alum elicited VAERD, indicated by Th2 driven immunity, reduced lung clearance, and increased lung pathology following challenge with a heterologous coronavirus. Replacement of alum with an alternative adjuvant reversed these properties leading to higher neutralizing antibody titers and clearance of the heterologous virus. The findings of this manuscript have important implications for human health and recipients of inactivated, alum-adjuvanted SARS-CoV-2 vaccines. These individuals could be at risk for VAERD following infection with distantly related coronaviruses. Given the implications for human health, the language in this manuscript should be carefully crafted to avoid citing unwarranted public health concerns that are not supported by the data.

Overall the conclusions from this manuscript are well-supported by the data and the appropriate controls are included. The findings of this manuscript represent an advancement in the field and suggests that VAERD should be monitored in current vaccine recipients and should be evaluated in pre-clinical studies as new coronavirus vaccine formulations are tested. The critiques outlined below are considered minor textual changes to increase clarity.

Critique:

The number of animals included within each experiment is not clear. From my reading, I believe the authors are using inappropriate statistical analysis when quantifying viral neutralization. The authors use experimental replicates of biological samples for the calculation of statistics. For example, Fig 1a represents data from 4, 12 and 19 animals per group (respectively). But this figure has 20-30 different data points within each group. Each point should be from a single animal and statistics should be derived from the data points of the individual animals. This same error is in 2A and Ext Fig 2. Possibly I am misunderstanding the data, but can the authors clarify if this is the case.

Figure 2. Can the authors specify the number of mice used in these experiments.

Figure 3. Can the authors specify the number of mice used in these experiments. Also is each dot representative of a single mouse in 3a?

Lines 208 and in methods. Please indicate the specific strain that is used for the inactivated SARS-CoV-2. This is a very important component of the manuscript.

Given the differential phenotype of the RIBI and alum adjuvants, can the authors provide more details on the adjuvant properties of the RIBI in the introduction or discussion. What is the specific molecule and what are the innate immune pathways invoked by this adjuvant? A general statement on Th1-driven immunity is not enough details.

Lines 414-416: The authors must be very careful in the wording of this section. As written, reading by the general public could infer that vaccinated individuals are more susceptible to circulating SARS-CoV-2 strains and drive further vaccine hesitancy. Yet the data ONLY demonstrates VAERD to a non-circulating distantly related SARS coronavirus. There is some evidence of Th2 immunity in the Beta MA10 challenge in Ext Fig 3, however lung pathology was lower than the controls. Abstract lines 101-103 could also be tweaked to reflect that the challenge strain Rs-SHC014-CoV which causes VAERD is not circulating in the human populations (... yet).

Reviewer #2:

Remarks to the Author:

Vaccine-associated enhanced respiratory disease (VAERD) has been an important issue in vaccine development against coronaviruses. This study will provide important insights for future COVID-19 vaccine policy considerations. Although the findings obtained in the mouse model have limitations, the importance of this study will be recognized as various mutant strains are now emerging.

The authors first described that challenge inoculation of mice immunized with an Alum adjuvant inactivated vaccine using the ancestral SARS-CoV-2 strain with a mouse-adapted MA10 strain caused eosinophil-associated VAERD by a Th2-shifted immune response. Subsequently, a similar phenomenon was demonstrated in viral challenge with a mouse-adapted strain derived from the B.1.351 mutant. Furthermore, the possibility of VAERD after heterologous infection with Rs-SHV014-CoV of Sarbecovirus Clade 3, considered a candidate for the next pandemic, was demonstrated using the mouse model. In all cases, the RIBI adjuvant instead of (or as a booster for) Alum adjuvant promoted viral clearance and avoided or improved the enhanced disease. Importantly, the authors showed that CD4⁺ helper T cells primarily contribute to VAERD in the mouse model.

The authors' study is solid and compelling, and the message is clear. However, some information is missing and needs to be added. The following are the main and minor comments.

1. In this study, VAERD was evaluated with multiple markers in a mouse model. Please discuss (or explain the significance of each) which method of evaluation is considered the appropriate marker in animal studies of VAERD. the EPX⁺ cell count is an objective marker for pathological evaluation, however, does not appear to be indicated in the results of some experiments (Fig. 2h, Fig. 5 d, Extended Data Fig. 6c). Additional information is needed as this marker may be an important objective marker. Also in Figure 2h, there is a large variation in viral titer among individuals in the iCoV2+ alum group. Was there a correlation between viral titer/viral antigen positive cell count, CD4⁺ positive cell infiltration, and/or eosinophil infiltration in this group?
2. The pathology evaluation score ALI system described neutrophils, but is this distinguished from eosinophils? If not, it would be "polymorphonuclear leukocytes".
3. In Extended Data Fig. 1, the Type 2 cytokine gene values are more variable in the iCoV+Alum group than in the other groups. What could be the reason for this?
4. The data that CD4⁺ cells are involved in VAERD was one of the key messages of this study; it would be good to show objective data on the number of CD4⁺ cells in Extended Data Fig. 5.
5. Please describe the activity of the RIBI adjuvant used in place of alum adjuvant. Also, please add another mouse model for VAERD evaluation of SARS-CoV-2 (Iwata-Yoshikawa and Shiwa et al., Science advances 2022 <https://www.science.org/doi/10.1126/sciadv.abh3827>) should be added to discuss possible adjuvant effects.
6. Please add whether VAERD of coronaviruses is currently reported in humans and the limitations in mouse models.

minor

1. Fig. 1a, there seems to be a significant difference between iCoV2+Alum group and iCoV2+RIBI ($p=0.0054$), if so, please specify in the text.
2. In Extended Data Fig. 3, Abbreviation 'EPX' should be defined at first appearance.
3. Please correct [iCoV] to [iCoV2] in the text or in the figure legend (e.g. Fig. 3 (b, c)).
4. Please provide data on the neutralizing antibody titer of the individual in Fig. 4b before the challenge inoculation, if available.

RESPONSE TO REVIEWERS

Introduction

We are pleased to submit this revised version of our manuscript “Adjuvant-dependent effects on the safety and efficacy of inactivated SARS-CoV-2 vaccines during heterologous infection by a SARS-related coronavirus” for publication by *Nature Communications*. We thank the reviewers both for their enthusiasm for the work described in the manuscript, and for their thoughtful critiques which we have used to further improve the manuscript. In the revised manuscript, we have responded to each reviewer concern. Herein we describe our response to each reviewer comment point-by-point and provide further relevant details including location of the respective edits (with tracked changes) within the revised manuscript.

Reviewer #1 concerns

- 1. The number of animals included within each experiment is not clear. From my reading, I believe the authors are using inappropriate statistical analysis when quantifying viral neutralization. The authors use experimental replicates of biological samples for the calculation of statistics. For example, Fig 1a represents data from 4, 12 and 19 animals per group (respectively). But this figure has 20-30 different data points within each group. Each point should be from a single animal and statistics should be derived from the data points of the individual animals. This same error is in 2A and Ext Fig 2. Possibly I am misunderstanding the data, but can the authors clarify if this is the case.**

We apologize for the confusion caused by our description of animal numbers in the figure legends and thank the reviewer for the opportunity to clarify this important point.

We can confirm that the appropriate statistical analysis is being performed when quantifying viral neutralization and that this confusion is due to incomplete listing of animal numbers in the figure legends. All data points in the viral neutralization figures represent individual animals, and no animal is represented by more than one individual data point. We only use biological replicates to perform our statistical analyses and do not use technical replicates (i.e., no pseudo-replication). The reason the number of animals used in the viral neutralization assays is higher than other experiments is that not all vaccinated animals are eventually challenged with virus (i.e., we have more vaccine immune serum samples compared to samples from challenged animals).

The number of animals used in each experiment have been added to all figure legends.

- 2. Figure 2. Can the authors specify the number of mice used in these experiments.**

The number of animals used in each experiment have been added to all figure legends.

- 3. Figure 3. Can the authors specify the number of mice used in these experiments. Also is each dot representative of a single mouse in 3a?**

Each individual data point in Fig. 3a represents a sample from an individual animal, and no animal is represented by more than one data point. The number of animals used in each experiment have been added to all figure legends.

- 4. Lines 208 and in methods. Please indicate the specific strain that is used for the inactivated SARS-CoV-2. This is a very important component of the manuscript.**

We agree with the reviewer that this is critical information and apologize for this omission of important details. The inactivated SARS-CoV-2 vaccine (iCoV2) was derived from a wild-type infectious clone of an early pandemic isolate of SARS-CoV-2, the D614 strain based on the WA1 sequence (Hou et al 2020 [reference 51]).

These details have been added to the Results (lines 191-193) and Methods (line 645).

- 5. Given the differential phenotype of the RIBI and alum adjuvants, can the authors provide more details on the adjuvant properties of the RIBI in the introduction or discussion. What is the specific molecule and what are the innate immune pathways invoked by this adjuvant? A general statement on Th1-driven immunity is not enough details.**

We thank the reviewer for drawing our attention to this omission. RIBI is an oil-in-water emulsion (2% squalene and Tween 80) containing monophosphoryl lipid A, a non-toxic analogue of lipopolysaccharide that stimulates TLR4, and synthetic trehalose dicorynomycolate, a low-toxicity derivative of mycobacterial cord factor trehalose-6,6-dimycolate that is thought to stimulate a C-type lectin expressed by macrophages known as the Mincle receptor.

These details have been added to the Introduction (lines 158-162). If needed, we would also be happy to add this information to the Discussion per the editor's or reviewer's preference.

- 6. Lines 414-416: The authors must be very careful in the wording of this section. As written, reading by the general public could infer that vaccinated individuals are more susceptible to circulating SARS-CoV-2 strains and drive further vaccine hesitancy. Yet the data ONLY demonstrates VAERD to a non-circulating distantly related SARS coronavirus. There is some evidence of Th2 immunity in the Beta MA10 challenge in Ext Fig 3, however lung pathology was lower than the controls. Abstract lines 101-103 could also be tweaked to reflect that the challenge strain Rs-SHC014-CoV which causes VAERD is not circulating in the human populations (... yet).**

We appreciate the reviewer's thoughtfulness regarding the potential misinterpretation of this passage as written. We believe that this is an extremely important consideration regarding public perception and interpretation of our findings. We agree that our findings do not support the conclusion that vaccinated individuals are at risk of VAERD from currently circulating Omicron subvariants. However, we do think that our findings support the rationale for further investigation into this question using both animal challenge models and surveillance of vaccinated human populations.

As advised by the reviewer, we have revised the Abstract (line 15) and Discussion (lines 504-512) to improve the clarity of these passages.

Reviewer #2 concerns

1. In this study, VAERD was evaluated with multiple markers in a mouse model. Please discuss (or explain the significance of each) which method of evaluation is considered the appropriate marker in animal studies of VAERD. the EPX+ cell count is an objective marker for pathological evaluation, however, does not appear to be indicated in the results of some experiments (Fig. 2h, Fig. 5 d, Extended Data Fig. 6c). Additional information is needed as this marker may be an important objective marker. Also in Figure 2h, there is a large variation in viral titer among individuals in the iCoV2+ alum group. Was there a correlation between viral titer/viral antigen positive cell count, CD4+ positive cell infiltration, and/or eosinophil infiltration in this group?

Regarding markers of VAERD, we appreciate the opportunity to clarify this important point. Pulmonary eosinophil infiltration (measured in this study through staining of eosinophil peroxidase [EPX]), is a commonly used objective marker for vaccine-associated type 2 inflammation (often referred to as “VAERD”) in animal models. However, in addition to this useful and established method for evaluating vaccine-associated type 2 inflammation, we believe that other objective markers of immunopathology and/or VAERD can supplement measurement of eosinophil infiltration to provide a more comprehensive understanding of VAERD in this model. The additional objective markers used in this study include delayed viral clearance (via plaque assay), decreased respiratory function (via whole body plethysmography), CD4⁺ cell infiltration (via immunohistochemistry, similar to eosinophil infiltration), and both acute lung injury (ALI) and diffuse alveolar damage (DAD), two complementary parameters of pulmonary pathology that are evaluated using a semi-quantitative scoring system by a blinded pathologist who is certified by the American Board of Veterinary Practitioners. Moreover, an additional benefit of our study is to illustrate the use of these supplementary VAERD markers for the scientific community, particularly other researchers who study VAERD.

We have added discussion of the various markers of VAERD in animal models in the Discussion (lines 465-469). Additionally, as requested, we have added eosinophil/EPX quantitation to all figures that had not previously contained this information. Please note that the EPX micrographs in Extended Data Fig. 6c and Extended Data Fig. 7 have also been quantified in Fig. 2h and Fig. 4e, respectively.

Finally, we conducted a correlation analysis involving the iCoV2 + Alum group referred to in Figure 2h to assess for correlation between viral lung titers, CD4⁺ cell infiltration, and eosinophil infiltration. To increase the comprehensiveness of our multivariate analysis, we also added ALI and DAD. We show the results of the correlation analysis below in a multivariable table showing pairwise Spearman *r* values **(a)** as well as a table showing pairwise multiplicity-adjusted P values (two-tailed) **(b)**.

We found a significant correlation ($\alpha = 0.05$, two-tailed) between lung titers and ALI ($r = 0.67$, $p = 0.011$). Although not statistically significant in this analysis based on our α threshold, we observed that lung titers might correlate moderately with CD4⁺ cell infiltration ($r = 0.50$, $p = 0.071$) and DAD ($r = 0.48$, $p = 0.085$). We found no evidence of a significant correlation between lung titers and eosinophil infiltration ($r = 0.27$, $p = 0.355$). In fact, eosinophil infiltration did not correlate significantly with any other parameters, which may be due in part to the fact that we observed a relatively low magnitude of eosinophil infiltration at this time point. We also found that CD4⁺ cell infiltration correlated significantly with ALI ($r = 0.72$, $p = 0.005$) and DAD ($r = 0.73$, $p = 0.004$). As expected, we also saw a strong correlation between ALI and DAD ($r = 0.83$, $p = 3.68e-4$).

We have added this information to the Results (lines 313-326) as well as in a new Extended Data figure (Extended Data Fig. 7).

a**b**

	Lung titers (log10 pfu)	EPX ⁺ cells per mm ²	CD4 ⁺ cells per mm ²	ALI	DAD
Lung titers (log10 pfu)		0.3548	0.0708	0.0111	0.0852
EPX ⁺ cells per mm ²	0.3548		0.1413	0.1535	0.4235
CD4 ⁺ cells per mm ²	0.0708	0.1413		0.0047	0.0038
ALI	0.0111	0.1535	0.0047		0.0004
DAD	0.0852	0.4235	0.0038	0.0004	

(a) Multivariable correlation matrix with pairwise Spearman r values.

(b) Pairwise multiplicity-adjusted P values (two-tailed).

2. The pathology evaluation score ALI system described neutrophils, but is this distinguished from eosinophils? If not, it would be "polymorphonuclear leukocytes".

The reviewer is correct that the ALI scoring system does not distinguish neutrophils from eosinophils.

We updated the pathology sub-section of the Methods to clarify that the ALI scoring system measures polymorphonuclear leukocytes (lines 744 and 755).

3. In Extended Data Fig. 1, the Type 2 cytokine gene values are more variable in the iCoV+Alum group than in the other groups. What could be the reason for this?

This is a great question for which we do not have a definitive answer, although we think there may be several reasons for this.

Aluminum hydroxide adjuvants (referred to as 'Alum' in our study) are known to cause type 2-biased immune responses, especially in BALB/c mice (which are used in this study). The consistently low

levels of type 2 cytokine gene expression observed in the iFLU + Alum (mock-vaccine) and iCoV2 + RIBI groups is unsurprising, as neither vaccine is expected to induce strong antigen-specific type 2 immune responses. Because neither of these vaccine induces strong type 2 immune responses against coronavirus challenge, we observe low levels of type 2 cytokine gene expression in all samples, which explains the low variability observed in these groups.

However, the reason for the high variability in the iCoV2 + Alum group is less clear but consistent with our observations of higher variability in infection outcomes observed in this group throughout the study. Our leading explanation is that animals are at different stages of immune responses to infection. The data in Extended Data Fig. 1 represent type 2 cytokine gene expression at 5 days post-infection (5 DPI) by mouse-adapted SARS-CoV-2 (SARS-CoV-2-MA10) (Extended Data Fig. 1a) or B.1.351-MA10 (Extended Data Fig. 1b). By 5 DPI, the iCoV2 + Alum group exhibits undetectable viral loads (measured by plaque assay) of either virus (SARS-CoV-2-MA10 in Fig. 1d and B.1.351-MA10 in Extended Data Fig. 3d). Therefore, the animals depicted in Extended Data Fig. 1 have likely recently cleared infection or are in the process of eliminating low levels of virus (below the limit of detection of the plaque assay) or viral antigen. Animals that were, at the time of tissue collection, still in the process of eliminating virus or viral antigen, or had only recently done so, may still exhibit relatively high levels of inflammation, and hence type 2 cytokine gene expression. In contrast, animals that had resolved the infection and cleared viral antigen relatively earlier may have had more time to return to low levels of inflammation including baseline levels of type 2 cytokine gene expression.

4. The data that CD4+ cells are involved in VAERD was one of the key messages of this study; it would be good to show objective data on the number of CD4+ cells in Extended Data Fig. 5.

We agree with the reviewer that it is important to show these objective quantified data. Therefore, we quantified all CD4-stained immunohistochemistry samples related to the representative micrographs presented in Extended Data Fig. 5b. The quantified data are presented below and in Extended Data Fig. 5c.

With these quantified data, we observe that during Rs-SHC014-CoV (SHC014) infection, iCoV2 + Alum-vaccinated animals exhibit significantly higher pulmonary CD4⁺ cells per mm² compared to iFLU + Alum-vaccinated animals at 2 days post-infection (DPI) ($p = 0.0110$), and compared to both iFLU + Alum- and iCoV2 + RIBI-vaccinated animals at 5 DPI ($p = <0.0001$ and 0.0002 , respectively).

Because the major differences are observed at 5 DPI, only the 5 DPI data are summarized in the text (lines 293 and 294), but if needed, we are happy to specify the 2 DPI information in the text as well per the editor's or reviewer's preference.

CD4⁺ cell infiltration during heterologous Rs-SHC014-CoV (SCH014) infection.

Individual data points represent biological replicates. Results were analyzed using two-way ANOVA with Tukey's multiple comparisons correction. Solid horizontal lines overlaying data represent group means. Error bars represent group standard deviation. Solid horizontal lines above data represent pairwise comparisons with multiplicity-adjusted P values (two-tailed).

- Please describe the activity of the RIBI adjuvant used in place of alum adjuvant. Also, please add another mouse model for VAERD evaluation of SARS-CoV-2 (Iwata-Yoshikawa and Shiwa et al., Science advances 2022 <https://www.science.org/doi/10.1126/sciadv.abh3827>) should be added to discuss possible adjuvant effects.

As requested, we have added details describing the characteristics and activity of RIBI to the Introduction (lines 158-162). We have also added a reference to the Iwata-Yoshikawa and Shiwa et al publication in the Introduction (lines 138 and 139), along with two other publications involving VAERD in animal models that were referenced in the original manuscript (lines 134-138). We have also added the Iwata-Yoshikawa and Shiwa et al reference to the Discussion to discuss the importance of adjuvants on vaccine outcomes in mouse models of VAERD (lines 595-602).

- Please add whether VAERD of coronaviruses is currently reported in humans and the limitations in mouse models.

We have added clarification that VAERD of coronaviruses has not been reported in humans to the Discussion (lines 473, 504 and 505), as well as a discussion of the limitations of mouse models (lines 580-588).

Minor

- Fig. 1a, there seems to be a significant difference between iCoV2+Alum group and iCoV2+RIBI (p=0.0054), if so, please specify in the text.

The reviewer is correct that there is a significant difference in neutralization titers against SARS-CoV-2 induced by iCoV2 + Alum and iCoV2 + RIBI represented in Figure 1a. We have specified this difference, including the magnitude of the average neutralization titers of each group, in the Results (lines 198 and 199).

2. In Extended Data Fig. 3, Abbreviation ‘EPX’ should be defined at first appearance.

As requested, the abbreviation ‘EPX’ has been defined at first appearance in Extended Data Fig. 3. Additionally, this abbreviation has been defined at first appearance in other relevant figures in the revised manuscript.

3. Please correct [iCoV] to [iCoV2] in the text or in the figure legend (e.g. Fig. 3 (b, c)).

As requested, ‘iCoV’ has been corrected to ‘iCoV2’ in Figure 3(b,c) in the revised manuscript.

4. Please provide data on the neutralizing antibody titer of the individual in Fig. 4b before the challenge inoculation, if available.

We agree with the reviewer that this is an important question. Therefore, to address this point, we performed additional neutralization assays on the post-heterologous boost (pre-challenge) samples from the studies described in Fig. 4b and Extended Data Fig. 8. As shown below, while we observed a significant increase in homologous neutralizing activity against SARS-CoV2 in the recombinant S2P + RIBI reboosted animals compared to the other vaccine groups (Fig. a below), we unfortunately found that the high background neutralization activity against SHC014 (which we also observed in Fig. 2a) was even more pronounced in the older animals used in the heterologous boost studies in Fig. 4, as illustrated in Fig. b (below) by the high non-specific neutralizing activity in the animals that received only iFLU + Alum for their prime, 1st boost, and 2nd boost. Therefore, we were unable to detect any anti-SHC014 neutralizing activity over background in these samples.

Per the editor’s or reviewer’s preference, we would be happy to include these additional data in the manuscript.

(a) $\log_{10} IC_{80}$ serum neutralization titers against homologous SARS-CoV-2 (D614G).

(b) $\log_{10} IC_{80}$ serum neutralization titers against heterologous SHC014.

Other revisions

1. We revised the manuscript to respond to editor requests, improve formatting, and comply with the reporting summary and the editorial policy checklist.
2. We updated the References to reflect additional references used to respond to reviewer comments and/or improve the manuscript.

3. To facilitate comparisons of effect sizes for eosinophil infiltration across different figures, we updated most EPX quantitation figures by setting the y-axis maximum to 500 EPX⁺ cells per mm². The exception to this is Fig. 6f because these mice were double-boosted and thus exhibit significantly higher EPX⁺ cell density compared to regularly vaccinated animals represented in all other relevant figures.
4. Miscellaneous small textual edits were made to improve the clarity of the manuscript.

Reviewers' Comments:

Reviewer #1:

Remarks to the Author:

Thank you for addressing all reviewers' concerns.

Reviewer #2:

Remarks to the Author:

I appreciate your thoughtful response to the peer review comments. I would like to comment on the following two points.

1. The results on the correlation of each marker in the evaluation of VEARD are very interesting. The authors found a low correlation between eosinophil infiltration and other markers. They discussed that eosinophil infiltration can be excluded as a marker for VEARD evaluation. On the other hand, this may be due in part to the relatively weak degree of eosinophil infiltration at the time of observation, as the authors noted in their response to the reviewers. Indeed, it is possible that eosinophils do not play a critical role of the pathology in this model (or may simply be a consequence), and other disease markers should be considered simultaneously when evaluating VEARD. However, given the many findings to date regarding eosinophil infiltration in VEARD, the decision to exclude eosinophils as an evaluation marker should be made with caution. Thus the reviewer recommends mentioning the time point at which eosinophil infiltration was observed in this experiment, as noted above.
2. Thank you for additional neutralization assays on the post-heterologous boost samples according to the minor comment 4. Please include these additional data in the manuscript as extended data.

RESPONSE TO REFEREES

Introduction

We are pleased to submit this revised version of our manuscript “Adjuvant-dependent impact of inactivated SARS-CoV-2 vaccines during heterologous infection by a SARS-related coronavirus” for publication by *Nature Communications*. We thank the reviewers both for their enthusiasm for the work described in the manuscript, and for their thoughtful comments which we have used to further improve the manuscript. Herein we describe our response to each reviewer comments point-by-point and provide further relevant details including location of the respective edits (with tracked changes) within the revised manuscript.

Reviewer #1:

Thank you for addressing all reviewers' concerns.

We would like to thank the reviewer again for their helpful responses to our manuscript.

Reviewer #2 (Remarks to the Author):

I appreciate your thoughtful response to the peer review comments. I would like to comment on the following two points.

- 1. The results on the correlation of each marker in the evaluation of VEARD are very interesting. The authors found a low correlation between eosinophil infiltration and other markers. They discussed that eosinophil infiltration can be excluded as a marker for VEARD evaluation. On the other hand, this may be due in part to the relatively weak degree of eosinophil infiltration at the time of observation, as the authors noted in their response to the reviewers. Indeed, it is possible that eosinophils do not play a critical role of the pathology in this model (or may simply be a consequence), and other disease markers should be considered simultaneously when evaluating VEARD. However, given the many findings to date regarding eosinophil infiltration in VEARD, the decision to exclude eosinophils as an evaluation marker should be made with caution. Thus the reviewer recommends mentioning the time point at which eosinophil infiltration was observed in this experiment, as noted above.**

We appreciate the reviewer’s response. We agree that eosinophil infiltration is a useful marker for evaluation of VAERD. Due to the fact that eosinophil infiltration is an established evaluation marker (as noted by the reviewer), we do not mean to imply that this marker should be excluded from VAERD evaluation, but simply that use of other markers of VAERD (i.e., in addition to eosinophil infiltration) will promote better understanding of the immunopathological consequences and mechanisms of VAERD. We agree with the reviewer that exclusion of eosinophil analysis should be made cautiously, and we believe that this is likely not the optimal choice in many circumstances.

Per the reviewer’s recommendation, we have noted the time point at which eosinophil infiltration was observed in this experiment (lines 233, 339, and 341-347). Additionally, we have modified the text to clarify our message and to communicate that eosinophils are still an important component of VAERD analysis (lines 341-348).

- 2. Thank you for additional neutralization assays on the post-heterologous boost samples according to the minor comment 4. Please include these additional data in the manuscript as supplementary information.**

As requested, we have included these additional neutralization assays in the manuscript as supplementary information (Supplementary Figure 9; described in lines 277-281 and 420-430).

Other revisions

1. We revised the manuscript to respond to editor requests and to comply with the reporting summary and author checklist.
2. We updated the Source Data file to reflect additional supplementary information as requested by reviewer #2 and exact p-values for pairwise comparisons from figures depicting body weights and whole body plethysmography.
3. Miscellaneous small textual edits were made to improve the clarity of the manuscript.